# ACCO: Accumulate While You Communicate for Communication-Overlapped Sharded LLM Training

**Adel Nabli**[1,2]    **Louis Fournier**[1*]    **Pierre Erbacher**[1*]    **Louis Serrano**[1]
**Eugene Belilovsky**[2]    **Edouard Oyallon**[1]
[1]Sorbonne Université, CNRS, ISIR, Paris - France
[2]Mila - Quebec AI Institute, Concordia University, Montréal - Québec
edouard.oyallon@cnrs.fr

## Abstract

Training LLMs relies on distributed implementations using multiple GPUs to compute gradients in parallel with sharded optimizers. However, synchronizing gradients in data parallel setups introduces communication overhead that grows with the number of workers, limiting parallelization efficiency. Local optimization algorithms reduce communications but incur high memory costs as they prevent optimizer state sharding, hindering scalability. To address this, we propose **AC**cumulate while **CO**mmunicate (ACCO), a memory-efficient optimization algorithm for distributed LLM training. By synchronizing delayed gradients while computing new ones, ACCO reduces GPU idle time and supports heterogeneous hardware. To mitigate the convergence issues caused by delayed updates, we introduce a novel technique ensuring training dynamics align with standard distributed optimization. Compared to ZeRO-1, our approach is significantly faster and scales effectively across heterogeneous hardware.

## 1 Introduction

Training Large Language Models (LLMs) with billions of parameters requires thousands of GPUs operating in parallel [71]. This process typically relies on distributed backpropagation [31] and gradient-based optimizers such as Adam [27] or AdamW [36]. However, distributed optimization at this scale is both memory- and communication-intensive. In standard data-parallel training, memory bottlenecks arise primarily from the optimizer's internal states, especially under mixed-precision training. Techniques like ZeRO [55] mitigate this by sharding states across workers. Due to limited GPU memory and the large size of modern models, large-scale LLM training frameworks must rely on such sharded partitioning [64, 58, 2]. In addition to memory constraints, communication overhead becomes a dominant performance bottleneck, as synchronizing gradients and optimizer states across GPUs can exceed the time spent on actual computation [50], a problem expected to persist even with future hardware advances [53]. The impact of communication is further amplified by the cluster's interconnect topology: effective sharding requires high-bandwidth links, and heterogeneous hardware or slow interconnects introduce straggler effects that slow down the entire system [15, 41].

To mitigate these issues, various communication-efficient distributed optimization algorithms have been proposed, particularly in settings with limited bandwidth such as Federated Learning. Local-update methods [66, 75, 40, 29, 14] reduce communication by splitting training into inner loops (local steps) and outer loops (synchronization steps). While this approach reduces frequency of communication, it introduces additional hyperparameters compared to standard training, scales poorly with the number of workers, and significantly increases memory requirements. For instance, the state-of-the-art CO2 method [69] requires memory overhead equivalent to four model copies—much more

---

[*]Equal contributions.

39th Conference on Neural Information Processing Systems (NeurIPS 2025).

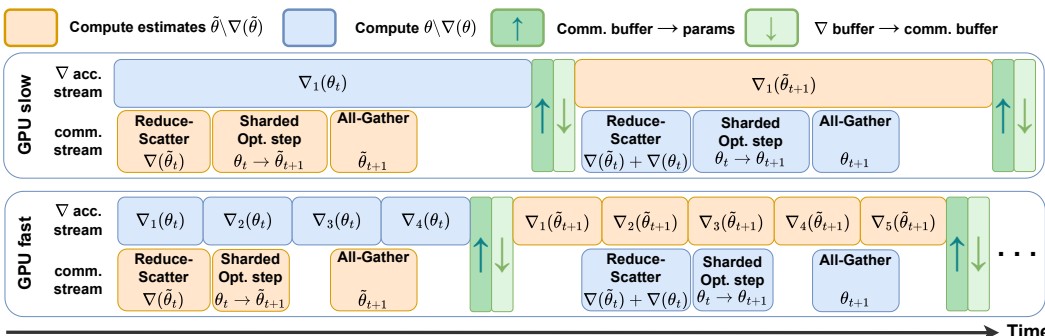

Figure 1: `ACCO` with a slow and a fast worker running in parallel, showing no idle time on both and hiding communications. The delayed update is compensated by splitting the mini-batch in two, leading to two steps in our timeline. The first uses half of the mini-batch to estimate "next step" parameters, and the second uses the full mini-batch to update them.

than standard distributed Adam and even further from its sharded variants [55]. Some local-update methods overlap communication with computation to hide latency, by executing both concurrently [74, 63, 82, 69]. However, this is challenging in sharded setups, since local updates typically require full optimizer states to be materialized, which defeats the purpose of memory or communication savings. A notable exception is ZeRO-Offload [60], which introduces the Delayed Parameter Update (DPU) mechanism: it runs gradient computation on GPUs while concurrently performing parameter updates on CPUs. Yet this approach suffers from one-step staleness—gradients [62] are applied to outdated parameters—which harms convergence [89]. Thus it struggles to match the performance or training dynamics of standard LLM training.

In this work, we introduce **AC**cumulate while you **CO**mmunicate (`ACCO`), a new optimization method that unifies the benefits of communication–computation overlap with memory-efficient training, matching the training dynamics of AdamW DDP without new hyperparameters to tune. Specifically **(1)** `ACCO` allows overlapping gradient computation and parameter updates in the memory-efficient sharded optimizer settings. **(2)** It eliminates the need for outer loops, reducing memory and tuning overhead. **(3)** Crucially, it compensates for the one-step delay introduced by parallel execution of communication and computation by using a novel accumulation mechanism, which avoids the convergence degradation observed in DPU-style updates. **(4)** Unlike prior approaches, `ACCO` is compatible with sharded training frameworks and requires no warm-up phase. **(5)** In the case of SGD, we prove that `ACCO` achieves the standard convergence rate, and **(6)** we confirm it empirically: `ACCO` consistently preserves convergence quality across both homogeneous and heterogeneous environments, with training curves that *mirror* those of AdamW—just as our theory predicts. **(7)** Our experiments on diverse LLM pretraining and fine-tuning tasks show that `ACCO` delivers substantial wall-clock speedups and effective communication hiding compared to ZeRO—without compromising training stability or memory efficiency. The code to reproduce all our experiments is available at `https://github.com/edouardoyallon/acco`.

## 2 Related work

**Local optimization methods for reducing communications.** Local optimization methods perform several local model updates between periodic averaging. With the SGD optimizer, these algorithms predate the deep learning era [90, 39], and their convergence properties are still investigated nowadays [87, 66, 76, 42]. Due to their practical and efficient communication scheme, they have since been used for the Distributed Training of Deep Neural Networks (DNNs) with methods such as EASGD [82], SlowMo [75] or Post-local SGD [34, 50], and are ubiquitous in Federated Learning [40, 29, 32], broadening the choice of optimizers beyond SGD [59, 25, 10]. They have also recently been applied in LLM training [14, 7]. Overlapping communications over consecutive steps of local computations allows to hide communication bottlenecks, resulting in algorithms such as Overlap local-SGD [74], COCO-SGD [63], or CO2 [69]. Moreover, with heterogeneous hardware, they can adapt their local computation rate to their hardware capacity [13, 38]. Yet, this comes at the price of additional memory requirements: due to their local nature [11], not only do these

Table 1: Comparisons of characteristics and memory consumption. $\Psi$: number of parameters in the model. $N$: number of workers. $K$: memory multiplier of the optimizer (Adam or AdamW). For SlowMo [75] and CO2 [69], no mention of mixed precision training is made. We assume they use it and that their additional terms are stored in half precision. While no additional momentum is required for our method, we still need a negligible communication buffer compared to the optimizers' states.

| Method | Overlap comm/comp | Hetero. hardware | No outer loop | Convergence Rates | Memory per replicas $(K, N, \Psi){=}(12, 64, 7.5\text{B})$ |
|---|---|---|---|---|---|
| DDP [31] | ✗ | ✗ | ✓ | ✓ | $(2{+}2{+}K)\Psi = 120$ GB |
| ZeRO-1 [55] | ✗ | ✗ | ✓ | ✓ | $(2{+}2{+}\frac{K}{N})\Psi = 31$ GB |
| ZeRO-2 [55] | ✗ | ✗ | ✓ | ✓ | $(2{+}\frac{2+K}{N})\Psi = 16$ GB |
| ZeRO-3 [55] | ✗ | ✗ | ✓ | ✓ | $(\frac{2+2+K}{N})\Psi = 2$ GB |
| SlowMo [75] | $\sim$ | ✗ | ✗ | $\sim$ | $(2{+}2{+}2{\times}2{+}K)\Psi = 150$ GB |
| DiLoCo [14] | ✓ | ✗ | ✗ | ? | $(2{+}2{+}2{\times}2{+}K)\Psi = 150$ GB |
| CO2 [69] | ✓ | ✗ | ✗ | ✓ | $(2{+}2{+}4{\times}2{+}K)\Psi = 180$ GB |
| DPU [60] | ✓ | ✗ | ✓ | ✗ | $(2{+}2{+}2{+}\frac{K}{N})\Psi = 46$ GB |
| WP [8] | ✓ | ✗ | ✓ | ✗ | $(2{+}2{+}2{+}\frac{K}{N})\Psi = 46$ GB |
| ACCO (Ours) | ✓ | ✓ | ✓ | ✓ | $(2{+}2{+}2{+}\frac{K}{N})\Psi = 46$ GB |

methods prevent the use of sharded optimizers such as ZeRO [55], but they also introduce additional control variables [75, 42, 69], hindering their scalability as shown in Tab. 1. Moreover, catering for heterogeneous hardware is not straightforward, as using different numbers of local updates leads to models shifting at different speeds, requiring extra care to counter this effect [38]. On the contrary, ACCO does not lead to such disparities. Since all devices share the same parameters, the device speeds difference only affects the mini-batch size computation. Similarly, doing multiple local optimizer updates makes the approach incompatible with optimizer sharding.

**Overlapping communications and computations.** For the asynchronous approaches, some approaches overlap gradient and communication steps, either explicitly [5], or by modeling them with independent stochastic processes [45, 44, 17]. However, none of these works focus on memory efficiency. Thus, they introduce additional variables and do not consider sharding the optimizer states. Moreover, they do not study optimizers other than SGD, and extending their beneficial properties to adaptive methods commonly used for DNN training such as Adam is still an ongoing research topic [4]. Delays being intrinsic to distributed asynchronous optimization, there is a rich literature studying them. In the case of distributed SGD in a parameter server setting, while early analysis showed convergence rates depending on the *maximal* delay [1, 67], recent lines of work improved these dependencies [28, 77, 19], proving that asynchronous SGD beats standard mini-batch SGD even with unbounded delays [41, 46]. However, they only study plain SGD, which is hardly used for DNN training. In this context, some work focused on the interplay between SGD with momentum and delays [43, 81], while delay compensation schemes such as re-scaling updates [86, 78] or buffering them [49] were proposed for Federated Learning. But still, they only study versions of SGD and not adaptive methods commonly used for LLMs training such as Adam [27] or AdamW [36]. Closer to our work, DPU was introduced as a memory-efficient way to train LLMs by running the optimizer on the CPU while gradients are computed on the GPU [60, 33], inducing a one-step delay between the gradients computed and the corresponding optimizer step. To mitigate it, they advise starting training by warming up for several steps with standard DDP. Perhaps surprisingly, we find in our experiments that this one-step delay has a noticeable influence on the convergence of LLMs training, even when using warmup steps. Contrary to DPU, we remove the need for them, with no impact on the convergence of our training. Moreover, as it is not its purpose, DPU still runs communications in the gradient computation stream, and is thus impacted both by the communication overhead of scaling and hardware heterogeneity. Finally, in pipeline parallelism, gradient delays also affect computation, and weight prediction methods have been proposed to mitigate their effect [8, 79, 30, 80].

**Memory-efficient distributed training of LLMs.** The activation memory overhead required for training Transformers [72] can be mitigated for an extra computational cost by reconstructing the input with reversible architectures [62, 23, 37], or recomputing the activations via checkpointing [9]. Efficient LLM training also combines parallelism methods. Classical data parallelism (DP) [12]

suffers both from a high communication volume and a linear increase in memory due to the model replicas. ZeRO-DP [56] and Fully-Sharded DP [85] avoid this issue by sharding the model states (i.e., the optimizer states, gradients, and parameters) between workers. This comes at the cost of further increasing the synchronization between workers and the communication volume, which can be mitigated by compression [73], memory trade-offs [83], or delayed gradients [88]. The memory can be even more reduced using expensive CPU-GPU communications to unload states on the CPU [60, 57]. On the other hand, model parallelism partitions the DNN components for parallelization, either with tensor parallelism [64] by slicing a layer's computation on several workers, or with pipeline parallelism, which divides a model into sets of layers trained in parallel on mini-batch slices. Popularized by [22], this method leaves some workers idling and an inefficient memory overhead [18]. Allowing delay in the gradients avoids worker idleness [47, 88] but exacerbates the memory overhead, which can be partially mitigated with gradient accumulation [48, 89] and activation checkpointing [26, 35]. Combining these frameworks results in the effective 3D parallelism [65].

## 3 Method

### 3.1 Background and Notations

We now present our framework as well as relevant prior methods for overlapping communication and computation from the perspective of an individual worker $i \in \{1, \ldots, N\}$. The goal is to minimize a differentiable objective function $f : \mathbb{R}^d \to \mathbb{R}$. All workers are initialized with identical parameters $\theta^{(0)} \in \mathbb{R}^d$, and at each iteration $t$, worker $i$ has access to a stochastic function $F : \mathbb{R}^d \times \Xi \to \mathbb{R}$, where $\Xi$ is a sample space derived from its local data shard. The gradient estimates are assumed to be unbiased: $\mathbb{E}[\nabla F(\theta, \xi)] = \nabla f(\theta)$ for $\xi \sim \Xi$. This setup allows for flexible, even time-varying, batch sizes depending on each machine's speed. However, for simplicity, we assume each worker computes a fixed-size minibatch of $N_i$ samples per iteration. The resulting local gradient estimate is given by

$$\nabla F_i(\theta, \xi) \triangleq \frac{1}{N_i} \sum_{k=1}^{N_i} \nabla F(\theta, \xi_k), \quad \text{where } \xi = (\xi_1, \ldots, \xi_{N_i}).$$

We also consider a generic optimizer, such as `Adam` or `AdamW` (common in LLM training), denoted by `Opt`. Applying the optimizer may require synchronizing internal states (e.g., moments, learning rates) across workers, which introduces a communication barrier. This synchronization can be particularly costly in settings involving optimizer sharding, and may substantially limit GPU throughput.

**Distributed Data Parallelism (DDP).** In a DDP setting, gradient computation and optimization steps are performed sequentially as follows, for a sequence of $\{\xi^{(t)}\}_t$:

$$g_i^{(t)} = \nabla F_i(\theta^{(t)}, \xi^{(t)}), \quad \theta^{(t+1)} = \texttt{Opt}\left(\theta^{(t)}, \sum_{i=1}^{N} \frac{N_i}{\sum_i N_i} g_i^{(t)}\right), \tag{DDP}$$

where gradients are averaged across all $N$ workers. As illustrated in this formulation, each step is fully synchronous and must follow a strict order, which limits the potential for overlapping communication and computation. A common strategy to address this limitation is to introduce two parameter buffers, denoted $\theta$ and $\tilde{\theta}$, where one is used for computation and the other for communication. Building on this insight, we next describe the main techniques used to achieve such overlap, and then `ACCO`.

**Delayed Parameter Update (DPU).** We describe the original DPU [60], and in our re-implementation, we run gradient communications in the same stream as the optimizer step, in parallel to the gradient computations. To prevent GPU from being idle at step $t$, gradients are accumulated over as many mini-batches as necessary until the communication process finishes. Then, DPU repeat the following, where each line can be run in parallel:

$$g_i^{(t+1)} = \nabla F_i(\tilde{\theta}^{(t)}, \xi^{(t)})$$
$$\tilde{\theta}^{(t+1)} = \theta^{(t)}, \quad \theta^{(t+1)} = \texttt{Opt}\left(\theta^{(t)}, \sum_{i=1}^{N} \frac{N_i}{\sum_i N_i} g_i^{(t)}\right). \tag{DPU}$$

Remark that, except at the first step $t = 0$, the gradients used by `Opt` are computed on parameters $\tilde{\theta}^{(t)} = \theta^{(t-1)}$ which differ from $\theta^{(t)}$, the ones we apply them to. This is inherently due to the parallel

nature of our execution, and what we denote by "delayed update". Sec. 4.3 shows that this has drastic impacts on the convergence, despite being only delayed by one time step. We hypothesize that, although the delay is limited to $\tau = 1$ and the dependency on delay is known to be linear [67], the training dynamics differ significantly from those of the standard setting.

**Weight Prediction (WP).**   Proposed by the work of [8] on mitigating pipeline delays, a simple estimation strategy is to reuse the most recently received gradients and apply a second optimizer step. Compared to DPU, this modifies the update rule for $\tilde{\theta}^{(t+1)}$, leading to:

$$g_i^{(t+1)} = \nabla F_i(\tilde{\theta}^{(t)}, \xi^{(t)})$$

$$\theta^{(t+1)} = \texttt{Opt}\left(\theta^{(t)}, \sum_{i=1}^N \frac{N_i}{\sum_i N_i} g_i^{(t)}\right), \quad \tilde{\theta}^{(t+1)} = \texttt{Opt}\left(\theta^{(t+1)}, \sum_{i=1}^N \frac{N_i}{\sum_i N_i} g_i^{(t)}\right). \tag{WP}$$

While this enables overlap by decoupling communication and computation, there is no formal guarantee that it leads to favorable convergence. We empirically evaluate this method in Sec. 4.3, and observe that its training dynamics deviate from the DDP baseline—unlike `ACCO`.

### 3.2   `ACCO`: a structured approach to Communication-Computation overlap.

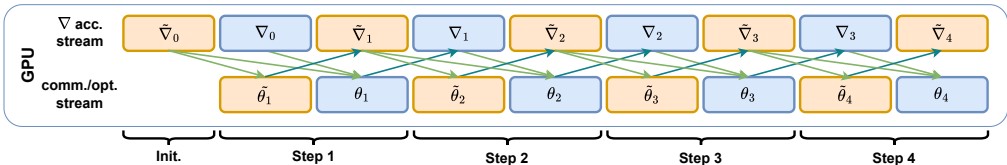

Figure 2: `ACCO`'s two-stage mechanism (1)-(2) to compensate the delayed updates via overlapping.

`ACCO`.   `ACCO` splits the computation of the mini-batch gradients into two successive stages, where the first half of the mini-batch is used to estimate $\tilde{\theta}^{(t+1)}$ while $\theta^{(t+1)}$ is calculated using the entire mini-batch. This is further motivated by the fact that gradient accumulation can be used to reach the extremely large batch sizes required to train LLMs [84], and if gradients are computed *sequentially* on a worker, we can leverage this to produce our estimate. Thus, the two stages, as in Fig. 2, are

$$(1)\begin{cases} g_i^{(t)} = \nabla F_i(\theta^{(t)}, \xi^{(t)}), \\ \tilde{\theta}^{(t+1)} = \texttt{Opt}\left(\theta^{(t)}, \sum_{i=1}^N \frac{N_i}{\sum_j N_j} \tilde{g}_i^{(t)}\right), \end{cases} (2)\begin{cases} \tilde{g}_i^{(t+1)} = \nabla F_i(\tilde{\theta}^{(t+1)}, \tilde{\xi}^{(t)}), \\ \theta^{(t+1)} = \texttt{Opt}\left(\theta^{(t)}, \sum_{i=1}^N \frac{N_i}{2\sum_j N_j}(g_i^{(t)} + \tilde{g}_i^{(t)})\right), \end{cases}$$

(ACCO)

We next describe the different components, whose streams can be run in parallel:

(1) The computation stream uses the second half of the mini-batch to compute the gradients $g_i^{(t)}$ with respect to parameters $\theta^{(t)}$ while the communication stream estimates what would be the next steps parameters $\tilde{\theta}^{(t+1)}$ using the estimated gradients $\tilde{g}_i^{(t)}$.

(2) The computation stream uses the first half of the mini-batch to estimate what would be the gradients $\tilde{g}_i^{(t+1)}$ of the next parameters $\theta^{(t+1)}$ using estimated parameters $\tilde{\theta}^{(t+1)}$ while the communication stream computes $\theta^{(t+1)}$ using the full mini-batch. Note that it starts from the same version of the parameters $\theta^{(t)}$ as in step (1). The first half $\tilde{g}_i^{(t)}$ was estimated at step (2) of the *last round*, while (1) compute the second half $g_i^{(t)}$.

**Theoretical analysis of `ACCO`.**   We now state our main results (SGD, for simplicity), with complete proofs provided in the appendix. The key idea underlying all proofs is that, for any minimizer $\theta^*$ of $f$ and any $\eta > 0$, the following function serves as a Lyapunov function for our dynamics

$$V(\theta, \tilde{\theta}) \triangleq f(\theta) - f(\theta^*) + \eta L(f(\tilde{\theta}) - f(\theta^*)) + L\|\theta - \tilde{\theta}\|^2 \geq 0.$$

**Proposition 3.1** (GD). *Let $f : \mathbb{R}^d \to \mathbb{R}$ be $L$-smooth and $\theta^* \in \arg\min f$. For $\eta \leq \frac{1}{2L}$, define*

$$\theta_{t+1} = \theta_t - \frac{\eta}{2}\left(\nabla f(\theta_t) + \nabla f(\tilde{\theta}_t)\right), \quad \tilde{\theta}_{t+1} = \theta_t - \eta \nabla f(\tilde{\theta}_t).$$

*Then, for any $T \geq 1$, and initializations $\theta_0, \tilde{\theta}_0 \in \mathbb{R}^d$, we have*

$$\frac{1}{T} \sum_{t=0}^{T-1} \left( \|\nabla f(\theta_t)\|^2 + \|\nabla f(\tilde{\theta}_t)\|^2 \right) \leq \frac{8}{\eta T} \left( f(\theta_0) + f(\tilde{\theta}_0) - 2f(\theta^*) + L\|\theta_0 - \tilde{\theta}_0\|^2 \right).$$

**Proposition 3.2** (SGD). *Under the same assumption as above, suppose $\theta_0 = \tilde{\theta}_0$ and we perform:*

$$\theta_{t+1} = \theta_t - \frac{\eta}{2}(g_t + \tilde{g}_t), \quad \tilde{\theta}_{t+1} = \theta_t - \eta \tilde{g}_t,$$

*where $g_t$ and $\tilde{g}_t$ are unbiased, conditionally independent estimators of $\nabla f(\theta_t)$ and $\nabla f(\tilde{\theta}_t)$, respectively, with bounded variance $\sigma^2$. Then, for $\eta \leq \frac{1}{2L}$ and any $T \geq 1$,*

$$\frac{1}{T} \sum_{t=0}^{T-1} \mathbb{E}\left[ \|\nabla f(\theta_t)\|^2 + \|\nabla f(\tilde{\theta}_t)\|^2 \right] \leq \frac{16}{\eta T}(f(\theta_0) - f(\theta^*)) + 8\sigma^2 L \eta.$$

We note that these rates recover the standard convergence guarantees of GD and SGD, unlike those in [75]. Indeed, unlike DPU or WP, `ACCO` does not rely on an approximation, which leads to a cleaner analysis and faster convergence, as reflected in our proof strategy. One can interpret DPU (with SGD as the optimizer `Opt`) as a parallel version of Delayed-SGD (D-SGD) with a one-step delay. While this setup has been shown to preserve asymptotic convergence rates in convex settings—such as quadratics [3] and strongly or quasi-convex functions [68]—our experiments (Sec. 4.3) show that, in practice, this delay significantly degrades performance when training LLMs with AdamW. In contrast, `ACCO` completely avoids delayed gradients, eliminating the impact of staleness. We note that the batch size in ACCO corresponds to the number of samples processed between two successive updates of $\theta$. Although it uses a pair of stochastic gradients, $\nabla F(\theta^{(t)}, \xi^{(t)})$ and $\nabla F(\tilde{\theta}^{(t)}, \tilde{\xi}^{(t)})$, both are computed synchronously with respect to $\theta^{(t)}$ (see Fig. 2). We confirm this advantage in Sec. 4, where `ACCO` yields training curves nearly indistinguishable from those of DDP (see Figs. 6, 5, 7).

## 4 Experiments

In this section, we present our experiments. Section 4.2 details the shared experimental setup. In Sec. 4.3, we demonstrate the shortcomings of DPU and WP—initially discussed in Sec. 3—which motivate the design of `ACCO`. This initial analysis focuses on small language models and datasets, using *TinyStories* [16] as a testbed. Sec. 4.4 shows that `ACCO` scales effectively by training a 125M-parameter GPT-Neo [6] on OpenWebText [21]. Sec. 4.5 pushes further with instruction tuning of a 2.7B GPT-Neo model, emphasizing communication bottle-

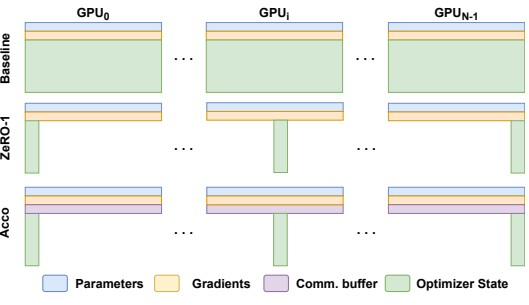

Figure 3: Memory requirements of `ACCO` vs DDP and ZeRO-1, see Tab.1 for quantitative details.

necks and the benefits of `ACCO`. Finally, Sec. 4.6 compares `ACCO` and DDP on heterogeneous hardware, where `ACCO` lets faster GPUs accumulate updates while waiting—unlike DDP—resulting in faster gradient computation.

### 4.1 Empirical motivation for `ACCO`

We first illustrate that the time spent communicating gradients can quickly trump the one used for computing them when using standard AdamW DDP to train LLMs. For that, we measure the time necessary to perform a full backward pass on a Llama-2 model [71] with 7B parameters hosted on a single GPU, using a batch size maxing out its memory. We compare this to the time necessary to compute an All-Reduce on those gradients with NCCL, varying the number of distributed workers. We experiment

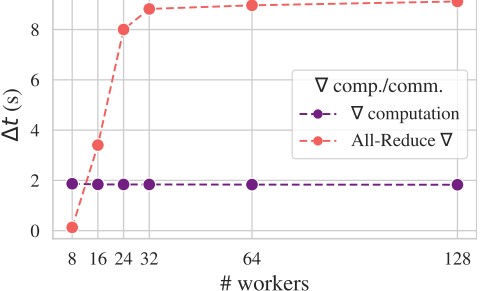

Figure 4: Time (per worker) spent computing and averaging gradients of a Llama-2 7B model for different numbers of GPUs.

on our local cluster of NVIDIA A100-80GB GPUs with 8 GPUs per node and an Omni-PAth inter-connection network at 100 Gb/s for inter-node connections, intra-node connections being done with NVLink 300 GB/s. Each worker is hosted on a single GPU. Fig. 4 shows that the communication time outside of a GPU node in our cluster to average the gradients across workers can take more than $4\times$ the one spent on the whole forward and backward step. As DDP only partially hides communications during the backward [31], this means that our GPUs remain idle the majority of the time when we use more than 24 distributed workers, motivating the need for methods leveraging this time to compute.

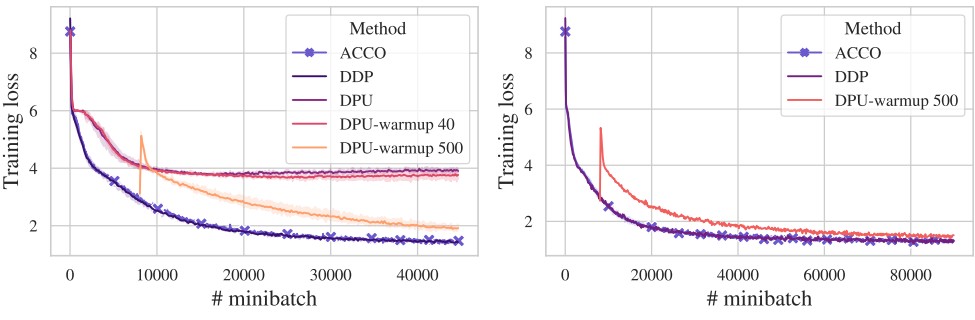

(a) Training with the specified amount in [16].    (b) Training for twice the specified amount.

Figure 5: Impact of the delayed update and warmup steps.

## 4.2 Experimental setup

Our experiments are performed on the GPU cluster described in Sec. 4.1. A detailed pseudo-code for `ACCO` can be found in App. B.2. Our code is in PyTorch [52], and we verified that our implementation produces two different streams running in parallel for the computations and communications using NVIDIA's Nsight System to profile it, shown in Fig. 13. We trained all our models with AdamW [36], using mixed precision: our model parameters, gradient accumulation and communication buffers are in `bfloat16` [24] while our sharded optimizer states are in single precision (see Fig. 3). As nowadays all distributed frameworks training LLMs at scale use a form of partitioning due to GPU memory constraints [58, 2], our main baseline is PyTorch's DDP [31] with ZeRO-1 [55] to shard the optimizer's state. As justified in Tab. 1, local optimization methods cannot be realistically considered for memory reasons. To compare in good faith DPU to `ACCO` in terms of wall-clock time, we also implemented our own version of DPU, as the implementation [61] solves a different problem than ours. The original ZeRO does not overlap computation and communication as it is designed to host the optimizer on the CPU, and is slower than ZeRO due to CPU and GPU memory transfers [60].

## 4.3 `ACCO` on TinyStories

We experiment with small language models on the TinyStories dataset [16], closely following their con-figuration and training hyperparameters. We use a 36M-parameter GPT-Neo-based [6] decoder-only transformer and train a BPE tokenizer on TinyStories to match their 10k vocabulary. All experiments are run with 8 workers on a single node.

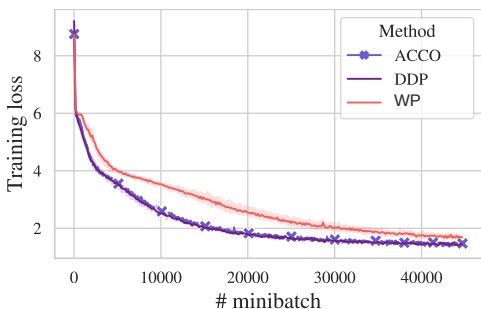

Figure 6: `ACCO` vs. WP on TinyStories

**Impact of delayed updates.** First, we investigate the impact of using delayed updates, re-purposing DPU [60] as described in Sec. 3. We run three vari-ants of this algorithm: **(1)** with no warmup, **(2)** with 40 warmup steps of non-delayed optimization step before switching to DPU (done in [60]), and **(3)** with 500 steps of warmup. We report in Fig. 5 our training losses on 8 distributed workers averaged over 3 runs. Using delayed updates greatly hurts convergence, especially when no or too few warmup steps are performed. Surprisingly, the number of warmup steps given in [60] does not work here, hinting that it is a sensitive hyper-parameter to tune for each use-case. If we train for twice as long as specified in [16], then the DPU training curve approaches the baseline one, without totally catching it.

Contrary to this, the training curve of our algorithm `ACCO` perfectly matches DDP, as advocated by our theory.

**Compensation via WP.** To mitigate the detrimental impact of using delayed updates, we test a first approach to mitigate it: WP as described in Sec. 3. This method applies two consecutive optimizer steps, re-using twice the same mini-batch. The first step produces the usual updated parameters, while the second predicts the parameters of the next step so that gradients can be computed on this estimate rather than on a stale version of the model. In Fig. 6, we compare the training curves of this delay-compensation method to ours. We remark that, while `ACCO` perfectly matches the DDP baseline at all times, WP displays worse behavior, especially at the beginning of the training. Thus, we dismiss this method and keep ours for the remaining experiments.

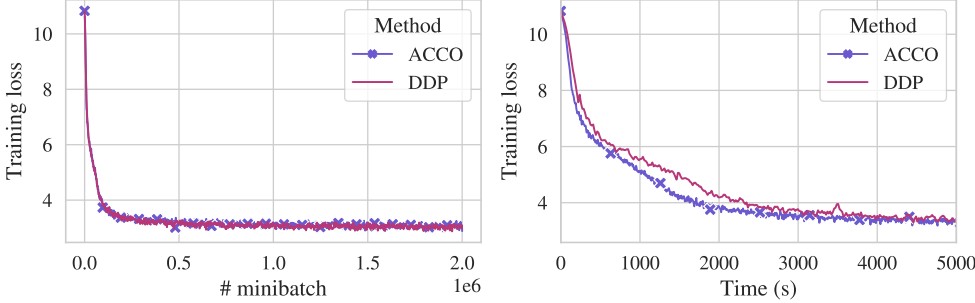

(a) Evolution of the loss over the whole training.  (b) Focus on the first part of the training w.r.t time.

Figure 7: Training curves for `ACCO` and DDP with 32 workers trained for 50B tokens.

### 4.4 Training GPT-Neo on OpenWebText

To assess how `ACCO` scales with larger models and more data, we pre-trained a model equivalent to GPT-2 [54] with both `ACCO` and DDP with a ZeRO optimizer. Specifically, we used the GPT-Neo architecture [6] with 125 million parameters and the OpenWebText dataset [21], which contains 40 GB of text. We used the GPT-Neo tokenizer, pre-trained on the Pile dataset [20]. The models were trained on sequences of 1024 tokens, with documents concatenated using end-of-sequence tokens. To assess the impact of using different hardware, the experiment was repeated on 2 different clusters. The first was conducted on 8 H100-PCIe 80GB on a single node. The second was on 32 A100-80G GPU distributed on 4 nodes. We maxed out the memory of our GPUs with a local mini-batch size of 24. To reach a sufficiently large overall batch size, we used 1 step of gradient accumulation for DDP, and none for `ACCO` as our method naturally accumulates over 1 step, resulting for the first and second experiments in respectively 400K and 1.5M tokens per effective batch for both `ACCO` and DDP. In Tab. 3, we report additional experimental details, and notice that training with `ACCO` allows for a 25% speedup on this pre-training task, which is additionally illustrated in Fig. 7. We also report that our implementation of `ACCO` adaptively scheduled 315 supplementary accumulation steps over the whole training to prevent GPUs from idling while waiting for communications.

Further details and results for the H100 experiment can be found in App. A. Tab. 2 reports the perplexity of trained language models with both methods. We evaluate the perplexity of language models on LAMBADA [51] and a test split of OpenWebText, and report similar results for both methods.

Table 2: Perplexity of our trained LLMs

| Method | LAMBADA (ppl ↓) | OpenWebText (ppl ↓) |
|---|---|---|
| `ACCO` 1x8 | 47.1 | 24.2 |
| DDP 1x8 | 47.5 | 24.3 |
| `ACCO` 4x8 | 45.5 | 22.5 |
| DDP 4x8 | 44.1 | 21.7 |

### 4.5 `ACCO` for instruction fine-tuning

In the former sections, we compared `ACCO` against DDP with ZeRO in the pre-training stage. To further validate our algorithm, we consider the GPT-Neo 2.7B model [6] pre-trained on the Pile dataset [20] and finetuned it on the Alpaca dataset [70] containing 52k pairs of instruction/answer. We fine-tuned the model using two configurations: 8 A100-80G on a single node, and 8 A100-80G distributed equally across 2 nodes. Samples are padded to match the longest sequence in the mini-

batch. We fixed the mini-batch size at 4, leading to a total batch size of 128 for all methods. For DDP and DPU, we used a gradient accumulation of 4, while for `ACCO`, a gradient accumulation of 2 to account for the `ACCO` accumulation described in Sec. 1. The learning rate was set to $2 \times 10^{-5}$, and with a warmup of 50 steps for DPU. In this setting, padding to the longest sequence in the mini-batch induces more variability in the number of tokens per mini-batch. This results in more variability in the computational load for each worker, leading to increased wait times for synchronization. We observe in Fig. 8 that `ACCO` achieves a low validation loss faster than DDP in both settings. Notably, the difference between `ACCO` DDP becomes more pronounced when workers are distributed across multiple nodes. Additionally, as shown in Tab. 3, larger models and optimizers result in longer communication times, further demonstrating the efficiency of `ACCO` in mitigating communication bottlenecks. This advantage translates to an 87% speedup for `ACCO` (see Tab. 3), highlighting the significant impact of communication bottlenecks on standard methods.

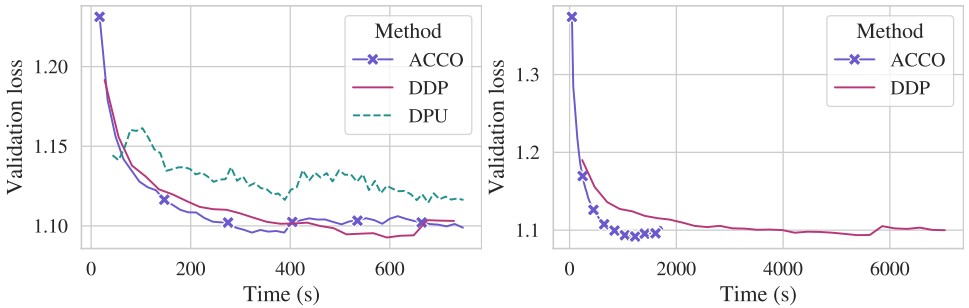

Figure 8: Validation curve with 8 workers on 1 node **(left)**, and 4 workers/node on 2 nodes **(right)**.

## 4.6 Experiment Using Heterogeneous Devices

To witness the impact of using heterogeneous devices, we run `ACCO` and compare it to DDP in a four workers setting, with one of the GPU four times slower than the other three. The training setting is the same as in Sec. 4.3. As we experiment on a A100 GPUs cluster, we simulate the heterogeneity of the hardware using the

Table 3: Pre-training (PT) and finetuning (FT) time speedup with `ACCO` against DDP on various setups with GPT-Neo.

| Stage | Model | GPUs | #tokens | ZeRO-1 | ACCO |
|-------|-------|------|---------|--------|------|
| **PT** | 125M | 1x8 | 6B | 4h41m | 4h25m |
| | | 4x8 | 50B | 14h41m | 10h55m |
| **FT** | 2.7B | 1x8 | 80M | 43min | 25min |
| | | 2x4 | 80M | 3h46m | 29min |

`time.sleep()` python command. First, we measure the time that a standard forward-backward step takes, and make one of the four GPUs idle for three times this amount after each forward-backward pass. In this context, DDP is only as fast as the slowest worker: 3 out of the 4 workers are idle 3/4 of the time. With `ACCO`, the other workers accumulate during the time they wait for the slow one to finish. Thus, `ACCO` allows to compute gradients for large batch sizes faster than standard baselines, resulting significantly smaller wall clock time, as shown in Fig. 9.

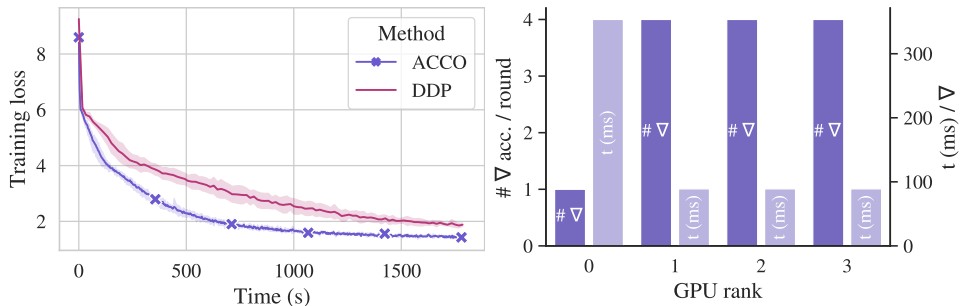

Figure 9: Training curves with 1 slow worker ($4\times$ slower).

## Conclusion

`ACCO` is a novel algorithm that addresses both memory and communication bottlenecks in distributed LLM training, with provable guarantees which match standard SGD. By overlapping gradient computation and communication while partitioning optimizer states, `ACCO` reduces communication overhead in a memory-efficient way. A new two-stage compensation mechanism corrects for delayed updates, ensuring convergence comparable to standard optimizers—without requiring warmup. Empirical results show significant speedups across pre-training and fine-tuning tasks, particularly in multi-node and heterogeneous environments.

## Acknowledgements

This work was supported by Project ANR-21-CE23-0030 ADONIS, EMERG-ADONIS from Alliance SU, PEPR IA (grant SHARP ANR-23-PEIA-0008), and the Sorbonne Center for Artificial Intelligence (SCAI) of Sorbonne University (IDEX SUPER 11-IDEX-0004). It was granted access to the AI resources of IDRIS under the allocation 2023-A0151014526 made by GENCI. EB acknowledges funding from FRQNT New Scholar and computational support from CFI.

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

## A    Experimental Details and Further Results

### A.1    Pre-training on TinyStories

For experiments in Sec. 4.3, we used the configuration available on the Huggingface Hub [1]. We trained our own 10k vocabulary tokenizer on the dataset. We also report in Fig. 10 the results of our study on the impact of halving the batch size for DPU by not performing any gradient accumulation (*i.e.*, performing an optimizer's step at each communication).

---

[1]Tiny Stories Available at: https://huggingface.co/datasets/roneneldan/TinyStories

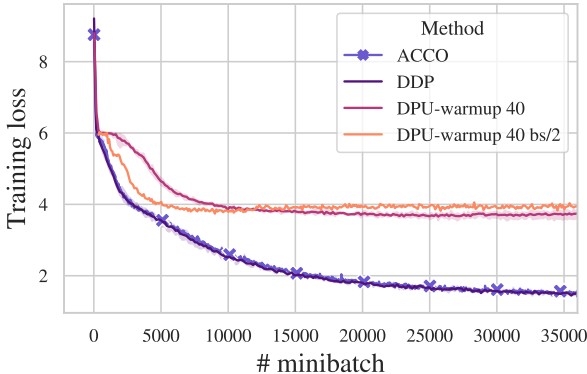

Figure 10: Comparison between running DPU on 8 GPUs with 2 steps of gradient accumulation on each (to match the standard batch-size) and DPU with only 1 gradient accumulation step. Doing so allows to double the number of optimizer's step per minibatch, but divides the effective batch size by 2. This leads to faster convergence early in the training, but worse training loss in the end.

## A.2    Proofs of Sec. 3

We first prove the convergence of ACCO in the Gradient Descent case.

**Proposition A.1** (Gradient Descent Case). *Let $f : \mathbb{R}^d \to \mathbb{R}$ be an L-smooth function, and consider the iterates defined by*

$$\theta_{t+1} = \theta_t - \frac{\eta}{2} \left( \nabla f(\theta_t) + \nabla f(\tilde{\theta}_t) \right), \quad \tilde{\theta}_{t+1} = \theta_t - \eta \nabla f(\tilde{\theta}_t),$$

*initialized at $\theta_0, \tilde{\theta}_0 \in \mathbb{R}^d$. Assume that $f$ admits a global minimizer $\theta^* \in \arg \min f$. Then, for any $T > 0$ and step size $\eta \leq \frac{1}{L}$, the following bound holds:*

$$\frac{1}{T} \sum_{t=0}^{T-1} \left( \|\nabla f(\theta_t)\|^2 + \|\nabla f(\tilde{\theta}_t)\|^2 \right) \leq \frac{8}{\eta T} \left( f(\theta_0) + f(\tilde{\theta}_0) - 2f(\theta^*) + L\|\theta_0 - \tilde{\theta}_0\|^2 \right).$$

*Proof.* We define the Lyapunov potential:

$$V(\theta, \tilde{\theta}) := (f(\theta) - f(\theta^*)) + \eta L \left( f(\tilde{\theta}) - f(\theta^*) \right) + L\|\theta - \tilde{\theta}\|^2.$$

Using the $L$-smoothness of $f$, we apply the standard descent lemma:

$$
\begin{aligned}
f(\theta_{t+1}) - f(\theta_t) &\leq -\frac{\eta}{2}\nabla f(\theta_t)^\top\left(\nabla f(\theta_t) + \nabla f(\tilde{\theta}_t)\right) + \frac{L\eta^2}{8}\left\|\nabla f(\theta_t) + \nabla f(\tilde{\theta}_t)\right\|^2 \\
&= -\frac{\eta}{2}\|\nabla f(\theta_t)\|^2 - \frac{\eta}{2}\langle\nabla f(\theta_t), \nabla f(\tilde{\theta}_t)\rangle \\
&\quad + \frac{L\eta^2}{8}\left(\|\nabla f(\theta_t)\|^2 + \|\nabla f(\tilde{\theta}_t)\|^2 + 2\langle\nabla f(\theta_t), \nabla f(\tilde{\theta}_t)\rangle\right) \\
&= \left(-\frac{\eta}{2} + \frac{L\eta^2}{8}\right)\|\nabla f(\theta_t)\|^2 + \frac{L\eta^2}{8}\|\nabla f(\tilde{\theta}_t)\|^2 + \left(-\frac{\eta}{2} + \frac{L\eta^2}{4}\right)\langle\nabla f(\theta_t), \nabla f(\tilde{\theta}_t)\rangle \\
&= \left(-\frac{\eta}{2} + \frac{L\eta^2}{8}\right)\|\nabla f(\theta_t)\|^2 + \frac{L\eta^2}{8}\|\nabla f(\tilde{\theta}_t)\|^2 \\
&\quad + \left(-\frac{\eta}{2} + \frac{L\eta^2}{4}\right)\cdot\frac{1}{2}\left(\|\nabla f(\theta_t)\|^2 + \|\nabla f(\tilde{\theta}_t)\|^2 - \|\nabla f(\theta_t) - \nabla f(\tilde{\theta}_t)\|^2\right) \\
&= \left(-\frac{\eta}{2} + \frac{L\eta^2}{8} + \frac{1}{2}\left(-\frac{\eta}{2} + \frac{L\eta^2}{4}\right)\right)\|\nabla f(\theta_t)\|^2 \\
&\quad + \left(\frac{L\eta^2}{8} + \frac{1}{2}\left(-\frac{\eta}{2} + \frac{L\eta^2}{4}\right)\right)\|\nabla f(\tilde{\theta}_t)\|^2 \\
&\quad - \frac{1}{2}\left(-\frac{\eta}{2} + \frac{L\eta^2}{4}\right)\|\nabla f(\theta_t) - \nabla f(\tilde{\theta}_t)\|^2 \\
&= \left(-\frac{3\eta}{4} + \frac{L\eta^2}{4}\right)\|\nabla f(\theta_t)\|^2 + \left(-\frac{\eta}{4} + \frac{L\eta^2}{4}\right)\|\nabla f(\tilde{\theta}_t)\|^2 \\
&\quad + \left(\frac{\eta}{4} - \frac{L\eta^2}{8}\right)\|\nabla f(\theta_t) - \nabla f(\tilde{\theta}_t)\|^2 \\
&\leq \left(-\frac{3\eta}{4} + \frac{L\eta^2}{4}\right)\|\nabla f(\theta_t)\|^2 + \left(-\frac{\eta}{4} + \frac{L\eta^2}{4}\right)\|\nabla f(\tilde{\theta}_t)\|^2 \\
&\quad + \left(\frac{\eta}{4} - \frac{L\eta^2}{8}\right)L^2\|\theta_t - \tilde{\theta}_t\|^2
\end{aligned}
$$

For the second point update, again using smoothness:

$$
\begin{aligned}
f(\tilde{\theta}_{t+1}) - f(\tilde{\theta}_t) &\leq \nabla f(\tilde{\theta}_t)^\top\left(\theta_t - \tilde{\theta}_t - \eta\nabla f(\tilde{\theta}_t)\right) + \frac{L}{2}\left\|\theta_t - \tilde{\theta}_t - \eta\nabla f(\tilde{\theta}_t)\right\|^2 \\
&= \frac{1}{2\eta}\left\|\theta_t - \tilde{\theta}_t\right\|^2 + \left(\frac{\eta}{2} - \eta\right)\left\|\nabla f(\tilde{\theta}_t)\right\|^2 + \left(\frac{L}{2} - \frac{1}{2\eta}\right)\left\|\theta_t - \tilde{\theta}_t - \eta\nabla f(\tilde{\theta}_t)\right\|^2 \\
&= \frac{1}{2\eta}\left\|\theta_t - \tilde{\theta}_t\right\|^2 + -\frac{\eta}{2}\left\|\nabla f(\tilde{\theta}_t)\right\|^2 + \left(\frac{L}{2} - \frac{1}{2\eta}\right)\left\|\theta_t - \tilde{\theta}_t - \eta\nabla f(\tilde{\theta}_t)\right\|^2
\end{aligned}
$$

The change in the regularization term satisfies, using $L$-smoothness:

$$
\|\theta_{t+1} - \tilde{\theta}_{t+1}\|^2 - \|\theta_t - \tilde{\theta}_t\|^2 = \frac{\eta^2}{4}\left\|\nabla f(\theta_t) - \nabla f(\tilde{\theta}_t)\right\|^2 - \left\|\theta_t - \tilde{\theta}_t\right\|^2 \leq \left(\frac{L^2\eta^2}{4} - 1\right)\left\|\theta_t - \tilde{\theta}_t\right\|^2.
$$

Thus, for the sake of readability if $0 \leq \eta \leq \frac{1}{2L}$, we have that:

$$
f(\theta_{t+1}) - f(\theta_t) \leq -\frac{1}{8}\eta\|\nabla f(\theta_t)\|^2 - \frac{1}{8}\eta\left\|\nabla f(\tilde{\theta}_t)\right\|^2 + \frac{L}{8}\left\|\theta_t - \tilde{\theta}_t\right\|^2 \tag{1}
$$

and

$$f(\tilde{\theta}_{t+1}) - f(\tilde{\theta}_t) \leq \frac{1}{2\eta} \|\theta_t - \tilde{\theta}_t\|^2$$

and

$$\|\theta_{t+1} - \tilde{\theta}_{t+1}\|^2 - \|\theta_t - \tilde{\theta}_t\|^2 \leq -\frac{3}{4} \|\theta_t - \tilde{\theta}_t\|^2$$

Combining the three terms (two function values and one regularizer), the total Lyapunov change is:

$$V(\theta_{t+1}, \tilde{\theta}_{t+1}) - V(\theta_t, \tilde{\theta}_t) \leq -\frac{1}{8}\eta \|\nabla f(\theta_t)\|^2 - \frac{1}{8}\eta \left\|\nabla f(\tilde{\theta}_t)\right\|^2 + (\frac{L}{8} + \frac{L}{2} - \frac{3L}{4}) \left\|\theta_t - \tilde{\theta}_t\right\|^2$$
$$\leq -\frac{1}{8}\eta \|\nabla f(\theta_t)\|^2 - \frac{1}{8}\eta \left\|\nabla f(\tilde{\theta}_t)\right\|^2$$

Summing over $t = 0$ to $T - 1$ and noting $V(\theta_T, \tilde{\theta}_T) \geq 0$, we conclude:

$$\sum_{t=0}^{T-1} \left( \|\nabla f(\theta_t)\|^2 + \|\nabla f(\tilde{\theta}_t)\|^2 \right) \leq \frac{4}{\eta}(V(\theta_0, \tilde{\theta}_0) - V(\theta_T, \tilde{\theta}_T)) \leq \frac{8}{\eta} \left( f(\theta_0) + f(\tilde{\theta}_0) - 2f(\theta^*) + L\|\theta_0 - \tilde{\theta}_0\|^2 \right).$$

Dividing both sides by $T$ gives the claimed result. $\qquad\square$

We now prove the convergence of ACCO in the Stochastic Gradient Descent case.

**Proposition A.2** (Stochastic Gradient Descent Case with Bounded Variance). *Let $f : \mathbb{R}^d \to \mathbb{R}$ be an $L$-smooth function, and let $\theta_0 = \tilde{\theta}_0 \in \mathbb{R}^d$ be the initialization. Suppose we perform the updates:*

$$\theta_{t+1} = \theta_t - \frac{\eta}{2}(g_t + \tilde{g}_t), \quad \tilde{\theta}_{t+1} = \theta_t - \eta\tilde{g}_t,$$

*where $g_t$ and $\tilde{g}_t$ are unbiased stochastic gradients of $f$, conditionally independent, at $\theta_t$ and $\tilde{\theta}_t$ respectively:*

$$\mathbb{E}[g_t \mid \theta_t, \tilde{\theta}_t] = \nabla f(\theta_t), \quad \mathbb{E}[\tilde{g}_t \mid \theta_t, \tilde{\theta}_t] = \nabla f(\tilde{\theta}_t),$$

*and assume the variance is bounded as*

$$\mathbb{E}\left[\|g_t - \nabla f(\theta_t)\|^2\right] \leq \sigma^2, \quad \mathbb{E}\left[\|\tilde{g}_t - \nabla f(\tilde{\theta}_t)\|^2\right] \leq \sigma^2.$$

*Then, for any $T > 0$ and step size $\eta \leq \frac{1}{2L}$, it holds that*

$$\frac{1}{T}\sum_{t=0}^{T-1} \mathbb{E}\left[\|\nabla f(\theta_t)\|^2 + \|\nabla f(\tilde{\theta}_t)\|^2\right] \leq \frac{16}{\eta T}(f(\theta_0) - f(\theta^*)) + 8\sigma^2 L\eta.$$

*Proof.* Using the same approach as in the full gradient case and $L$-smoothness, we get:

$$\mathbb{E}\left[f(\theta_{t+1}) \mid \theta_t, \tilde{\theta}_t\right] \leq f(\theta_t) - \frac{\eta}{2}\nabla f(\theta_t)^\top \mathbb{E}[g_t + \tilde{g}_t \mid \theta_t, \tilde{\theta}_t] + \frac{L\eta^2}{8}\mathbb{E}\left[\|g_t + \tilde{g}_t\|^2 \mid \theta_t, \tilde{\theta}_t\right],$$

$$\mathbb{E}\left[f(\tilde{\theta}_{t+1}) \mid \theta_t, \tilde{\theta}_t\right] \leq f(\tilde{\theta}_t) + \nabla f(\tilde{\theta}_t)^\top (\theta_t - \tilde{\theta}_t - \eta\mathbb{E}[\tilde{g}_t \mid \theta_t, \tilde{\theta}_t]) + \frac{L}{2}\mathbb{E}\left[\|\theta_t - \tilde{\theta}_t - \eta\tilde{g}_t\|^2 \mid \theta_t, \tilde{\theta}_t\right].$$

We also expand the expected change in the quadratic term:

$$\mathbb{E}\left[\|\theta_{t+1} - \tilde{\theta}_{t+1}\|^2 \mid \theta_t, \tilde{\theta}_t\right] = \mathbb{E}\left[\left\|\frac{\eta}{2}(g_t - \tilde{g}_t) + (\theta_t - \tilde{\theta}_t)\right\|^2 \mid \theta_t, \tilde{\theta}_t\right]$$
$$= \|\theta_t - \tilde{\theta}_t\|^2 + \frac{\eta^2}{4}\mathbb{E}\left[\|g_t - \tilde{g}_t\|^2 \mid \theta_t, \tilde{\theta}_t\right] + \eta\langle\theta_t - \tilde{\theta}_t, \mathbb{E}\left[g_t - \tilde{g}_t \mid \theta_t, \tilde{\theta}_t\right]\rangle.$$

Further simplufications give

$$\mathbb{E}\left[\|\theta_t - \tilde{\theta}_t - \eta\tilde{g}_t\|^2 \mid \theta_t, \tilde{\theta}_t\right] = \|\theta_t - \tilde{\theta}_t - \eta\nabla f(\tilde{\theta}_t)\|^2 + \eta^2 \operatorname{Var}(\tilde{g}_t \mid \tilde{\theta}_t),$$

$$\mathbb{E}\left[\|g_t + \tilde{g}_t\|^2 \mid \theta_t, \tilde{\theta}_t\right] = \|\nabla f(\theta_t) + \nabla f(\tilde{\theta}_t)\|^2 + \operatorname{Var}(g_t \mid \theta_t) + \operatorname{Var}(\tilde{g}_t \mid \tilde{\theta}_t),$$

$$\mathbb{E}\left[\|g_t - \tilde{g}_t\|^2 \mid \theta_t, \tilde{\theta}_t\right] = \|\nabla f(\theta_t) - \nabla f(\tilde{\theta}_t)\|^2 + \operatorname{Var}(g_t \mid \theta_t) + \operatorname{Var}(\tilde{g}_t \mid \tilde{\theta}_t).$$

Using the bounded variance a we can follow the same derivation as in the full-gradient case, with extra $\sigma^2$ terms appearing.

This yields:

$$\mathbb{E}\left[V(\theta_{t+1}, \tilde{\theta}_{t+1}) \mid \theta_t, \tilde{\theta}_t\right] - V(\theta_t, \tilde{\theta}_t) \leq -\frac{1}{8}\eta \|\nabla f(\theta_t)\|^2 - \frac{1}{8}\eta \left\|\nabla f(\tilde{\theta}_t)\right\|^2$$
$$+ \sigma^2 \left(\frac{L\eta^2}{4} + \frac{L^2\eta^3}{2} + \frac{L\eta^2}{4}\right).$$

Taking expectations and summing over $t = 0$ to $T-1$, then rearranging and using $0 \leq \eta T \leq \frac{1}{2}$ and $\theta_0 = \tilde{\theta}_0$, we obtain:

$$\frac{1}{T}\sum_{t=0}^{T-1} \mathbb{E}\left[\|\nabla f(\theta_t)\|^2 + \|\nabla f(\tilde{\theta}_t)\|^2\right] \leq \frac{16}{\eta T}(f(\theta_0) - f(\theta^*)) + 8\sigma^2 L\eta.$$

$\square$

Note that it would be possible to derive bounds for non-increasing step size, as $V(\theta, \tilde{\theta}, \eta) := (f(\theta) - f(\theta^*)) + \eta L\left(f(\tilde{\theta}) - f(\theta^*)\right) + L\|\theta - \tilde{\theta}\|^2$ satisfies $V(\theta, \tilde{\theta}, \eta) \leq V(\theta, \tilde{\theta}, \tilde{\eta})$ for $\eta \leq \tilde{\eta}$.

### A.3 Pre-training on OpenWebText

For all pre-training experiments on OpenWebText, the configuration used to instantiate the GPTNeo 125M is available on the Huggingface Hub[2]. We only changed the "max_position_embeddings" parameter from 2048 to 1024. More details are displayed in Tab. 4. We used the OpenWebText dataset available on Huggingface[3]. We also report in Fig. 11 further results for the pre-training on H100 GPUs.

### A.4 Instruction Fine-Tuning

For all fine-tuning experiments, we used the pre-trained GPT-neo 2.7B available on the Huggingface Hub[4] and the associated tokenizer. We used the Alpaca dataset available on Huggingface[5]. More details are displayed in Tab. 5.We also report in Fig. 12 further results on the experiment described in Sec. 4.5.

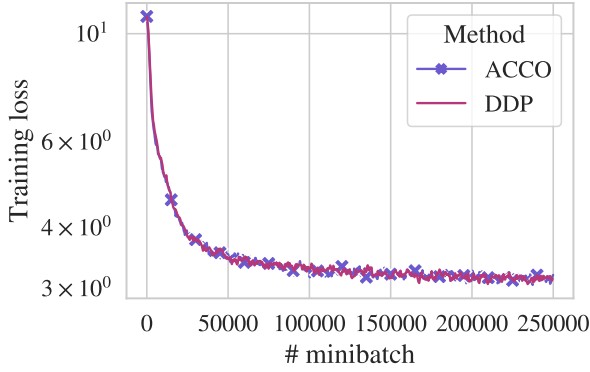

Figure 11: Training loss during training on OpenWebText with 8 H100 GPUs and 6B tokens.

Table 4: Training hyperparameters for ACCO and DDP configurations.

| Hyperparameter | 8 H100 | 32 A100 |
|---|---|---|
| mini-batch_size | 24 | 24 |
| n_grad_accumulation | ACCO: -DDP: 1 | ACCO: -DDP: 1 |
| sequence_len | 1024 | 1024 |
| #tokens_batch | 400K | 1.5M |
| optimizer | AdamW | AdamW |
| learning_rate | 6e-4 | 6e-4 |
| weight_decay | 0.1 | 0.1 |
| adam_beta1 | 0.9 | 0.9 |
| adam_beta2 | 0.95 | 0.95 |
| nb_steps_tot | 50000 | 50000 |
| scheduler | cosine | cosine |
| n_warmup_steps | 0 | 0 |

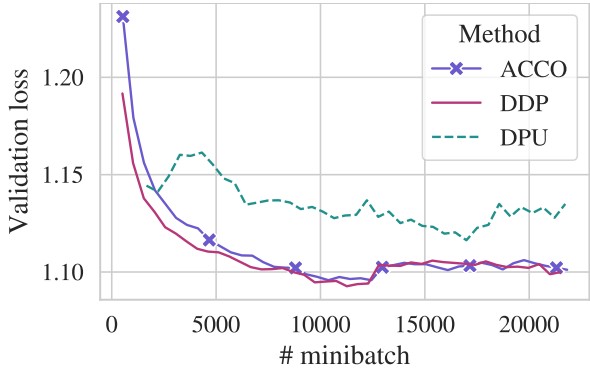

Figure 12: Validation curve with 8 workers on a single node.

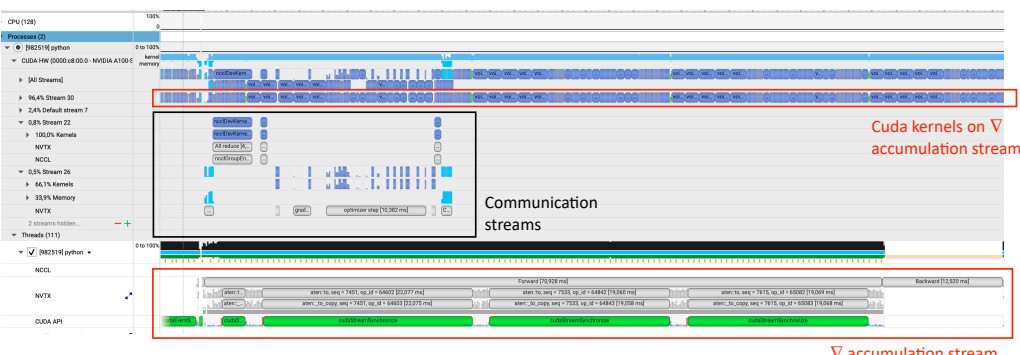

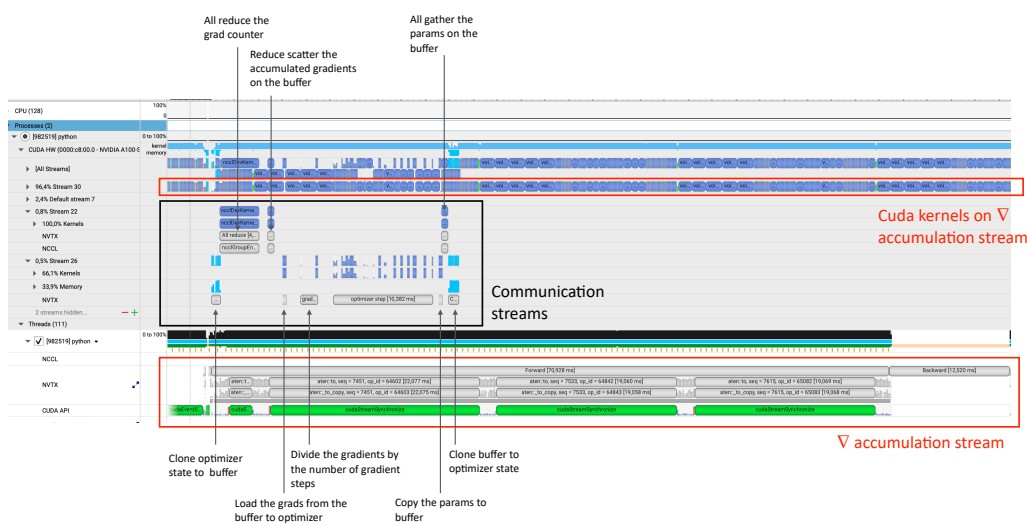

Figure 13: Nsight system profile of our implementation of ACCO: our two steams do run in parallel. In this Figure, the computation take more time than the communication because we only profiled small scale experiments with 8 workers, and small number of parameters (36M as we profiled on our TinyStories [16] setting). This changes when using larger models and more workers, as seen in 4.1.

Table 5: Finetuning hyperparameters for ACCO, DDP and DPU configurations.

| Hyperparameter | ACCO | DDP | DPU |
|---|---|---|---|
| mini-batch_size | 4 | 4 | 4 |
| n_grad_accumulation | 2 | 4 | 4 |
| total batch_size | 128 | 128 | 128 |
| optimizer | AdamW | AdamW | AdamW |
| learning_rate | 2e-5 | 2e-5 | 2e-5 |
| weight_decay | 0.0 | 0.0 | 0.0 |
| adam_beta1 | 0.9 | 0.9 | 0.9 |
| adam_beta2 | 0.95 | 0.95 | 0.95 |
| nb_steps_tot | 50000 | 50000 | 50000 |
| scheduler | cosine | cosine | cosine |
| n_warmup_steps | 0 | 0 | 50 |

# B  Implementation Details

## B.1  Profiling Results

## B.2  Algorithm Pseudo-Code

We present our algorithm for time-varying batch size $N_i^{(t)}$.

---

[2]GPT-neo 125M Configuration Available at: `https://huggingface.co/EleutherAI/gpt-neo-125m/blob/main/config.json`

[3]OpenWebText Dataset Available at: `https://huggingface.co/datasets/Skylion007/openwebtext`

[4]GPT-neo 2.7B Available at: `https://huggingface.co/EleutherAI/gpt-neo-2.7B`

[5]Alpaca Dataset Available at: `https://huggingface.co/datasets/tatsu-lab/alpaca`

## B.3 Slurm script to reproduce our results

Listing 1: SLURM and ACCO fine-tuning configuration

```
#SBATCH --nodes=2 # Request 2 nodes
#SBATCH --gres=gpu:1 # 1 GPU per node
#SBATCH --ntasks-per-node=1 # 1 task per node

acco-ft:
  group_by_length: false
  batch_size: 4
  n_grad_accumulation: 4
  learning_rate: 1e-5
  weight_decay: 0.0
  adam_beta1: 0.9
  adam_beta2: 0.95
  nb_steps_tot: 50000
  dataloader_num_workers: 1
  dataloader_pin_memory: True
  dataloader_persistent_workers: True
  label_smoothing_factor: 0
  max_length: 512
  scheduler_name: 'cosine'
  warmup: 0
  save: False
  use_mixed_precision: True
  n_warmup_steps: 0
  run_baseline_ddp: False # True for DDP
  method_name: 'acco' # 'ddp' for DDP
  #gradient_accumulation_steps: 1 # Add for DDP
  eval: True
  eval_step: 10
  run_expe_slow: False
  const_len_batch: False
  finetune: True
```

---

**Algorithm 1** Training with `ACCO` in parallel for a worker $i$

---

1: **Input:** Model with differentiable loss $F$, number of models $N$, initial parameters $\theta^{(0)}$, training steps $T$, dataset shards $\mathcal{D}_i$.

2: **Initialize:** gradients $g_i^{(-1)} = \nabla F(\theta^{(0)}, \xi_i^{(0)})$ and number of gradients $N_i^{(-1)} = 1$

3: **# Computation stream**

4: **while** $t < T$ **do**

5:     **Stage 1.**

6:     **while** not `Ready_for_Stage_2` **do**

7:         $\xi_i^{(t)} \leftarrow \mathcal{D}_i$

8:         $g_i^{(t)} \leftarrow g_i^{(t)} + \nabla F(\theta^{(t)}, \xi_i^{(t)})$

9:         $N_i^{(t)} \leftarrow N_i^{(t)} + 1$

10:     $\tilde{\theta}^{(t+1)} \leftarrow \textbf{Buffer}_i$

11:     $\textbf{Buffer}_i \leftarrow (N_i^{(t)}, g_i^{(t)})$

12:     **Stage 2.**

13:     **while** not `Ready_for_Stage_1` **do**

14:         $\xi_i^{(t)} \leftarrow \mathcal{D}_i$

15:         $\tilde{g}_i^{(t)} \leftarrow \tilde{g}_i^{(t)} + \nabla F(\tilde{\theta}^{(t+1)}, \xi_i^{(t)})$

16:         $\tilde{N}_i^{(t)} \leftarrow \tilde{N}_i^{(t)} + 1$

17:         $t \leftarrow t + 1$

18:     $\theta^{(t+1)} \leftarrow \textbf{Buffer}_i$

19:     $\textbf{Buffer}_i \leftarrow (\tilde{N}_i^{(t)}, \tilde{g}_i^{(t)})$

20:

21: **# Communication stream**

22: **while True do**

23:     **Stage 1.**

24:     $(\tilde{N}_i^{(t)}, \tilde{g}_i^{(t)}) \leftarrow \textbf{Buffer}_i$

25:     $\sum_i \tilde{N}_i^{(t)} \leftarrow \texttt{All\_Reduce}(\tilde{N}_i^{(t)})$

26:     $\texttt{Shard}_i\left(\sum_i g_i^{(t)}\right) \leftarrow \texttt{Reduce\_Scatter}(\tilde{g}_i^{(t)})$

27:     $\texttt{Shard}_i\left(\tilde{\theta}^{(t+1)}\right) \leftarrow \texttt{ShardedOpt}\left(\texttt{Shard}_i\left(\theta^{(t)}\right), \texttt{Shard}_i\left(\frac{\sum_i \tilde{g}_i^{(t)}}{\sum_i \tilde{N}_i^{(t)}}\right)\right)$

28:     $\textbf{Buffer}_i \leftarrow \texttt{All\_Gather}(\texttt{Shard}_i\left(\tilde{\theta}^{(t+1)}\right))$

29:     $N_i^{(t)} \leftarrow 0$

30:     `Ready_for_Stage_2` $\leftarrow$ **True**

31:     `Ready_for_Stage_1` $\leftarrow$ **False**

32:     **Stage 2.**

33:     $(N_i^{(t)}, g_i^{(t)}) \leftarrow \textbf{Buffer}_i$

34:     $\sum_i N_i^{(t)} + \tilde{N}_i^{(t)} \leftarrow \texttt{All\_Reduce}(N_i^{(t)} + \sum_i \tilde{N}_i^{(t)})$

35:     $\texttt{Shard}_i\left(\sum_i g_i^{(t)} + \tilde{g}_i^{(t)}\right) \leftarrow \texttt{Reduce\_Scatter}(g_i^{(t)} + \sum_i \tilde{g}_i^{(t)})$

36:     $\texttt{Shard}_i\left(\theta^{(t+1)}\right) \leftarrow \texttt{ShardedOpt}\left(\texttt{Shard}_i\left(\theta^{(t)}\right), \texttt{Shard}_i\left(\frac{\sum_i g_i^{(t)} + \tilde{g}_i^{(t)}}{\sum_i N_i^{(t)} + \tilde{N}_i^{(t)}}\right)\right)$

37:     $\textbf{Buffer}_i \leftarrow \texttt{All\_Gather}(\texttt{Shard}_i\left(\theta^{(t+1)}\right))$

38:     $\tilde{N}_i^{(t)} \leftarrow 0$

39:     `Ready_for_Stage_1` $\leftarrow$ **True**

40:     `Ready_for_Stage_2` $\leftarrow$ **False**

---

