# OpenReview forum: "ACCO: Accumulate While You Communicate for Communication-Overlapped Sharded LLM Training"
_NeurIPS.cc/2025/Conference — NeurIPS 2025 poster_

### Official Review · Reviewer_E3rM · 2025-06-19

**Clarity:** 1
**Significance:** 2
**Originality:** 2
**Rating:** 4
**Confidence:** 4

**Summary:**

This work focus on distributed training using a fast workers of smaller batch size to estimate the "future weights" while the slow worker finishes gradient computation of the rest of the batch. This creates an opportunity to overlap communication and computation leaving no GPU idles time. In effect, the model is updated by gradients from the "estimated weights" and the gradient of "real weight", which in empirical studies show no obvious degradation from baseline DDP.

**Questions:**

The paper shows theoretical results only for GD and SGD. I wonder how this half-delayed gradient would impact the Adam's convergence, especially the how it would impact the estimation of first and second order momentums?

Since all experiments are done with Adam, without this the paper seems to be incomplete.

**Ethical Concerns:**

["NO or VERY MINOR ethics concerns only"]

**Final Justification:**

Overall, I believe this work is still borderline, due to following reasons: 1) the lack of comparison with state of the art methods (only comparing with methods that are almost half a decade old which lack large scale verifications despite of enough time has passed), not really providing assurance on update steps being close to the baseline (only showing semi-close training curve for specific cases, though showing a comparison on toy during rebuttal) and 3) writing and annotations are quite confusing enough for non-system background audience, which are most of our community in Neurips.

 I don't mind if it gets accepted, given some of our peers seem to really think highly of this work. But my opinion remains that has not clearly breach the threshold for acceptance by a large margin.

**Limitations:**

Limitation has been discussed and addressed

**Paper Formatting Concerns:**

No formatting issues have been noticed

**Quality:**

2

**Strengths And Weaknesses:**

# Strengths
* Wall-clock measurement and solid experiments on LLM pretraining
* The idea is intuitive and well-motivated

# Weaknesses
* **Only comparing with naive baseline DPU.** DiLoco despite of seemingly taking more memory on paper, the extra memory can be easily offloaded, since it's only used every few hundred steps, and the cost can be negligible. A more recent version on Streaming DiLoco [1] avoids that issue completely and hiding latency by overlapping completely.

* **The description of algorithm is very confusing.** might be easier to just do with a chart or pesudo code blocks. Here is an example:
   "DPU repeat the following, where each line can be run in parallel ..." There is a read from $\tilde{\theta}^{(t)}$ for gradient computation and then there is an assignment to $\tilde{\theta}^{(t+1)}$ in the second line, if they are using the same buffer, it would cause a race condition. Similarly, the same thing happens to $g$. This makes me question the baseline's correctness.

* **Not enough ablation studies.**
What happens if it has different mini-batch size? This method intuitively should be highly sensitive to variation in mini-batch size, due to using only half of the batch to estimate the "future model weights". What happens, as quite a common thing nowadays, the per node batch size could be very small for large model training, in the extreme case, the batch size in the fast stream would go down to 1, which then would introduce large variance.

[1] Douillard, Arthur, et al. "Streaming DiLoCo with overlapping communication: Towards a Distributed Free Lunch." arXiv preprint arXiv:2501.18512 (2025).



Minor issue:
The all-reduce expression is unnecessarily complex. Just write out all-reduce wouldn't have been clearer

---

> ### Author Rebuttal · Authors · 2025-07-30
>
> We thank the reviewer for their comments. We’re glad that the **motivation and intuition** behind ACCO came through clearly, and we appreciate your recognition of the **"solid"** empirical results and wall-clock improvements demonstrated in our LLM pretraining experiments. We address your points and suggestions in detail below.
>
> ---
>
> **W1: “Only comparing to DPU”**
>
> We would like to clarify that in Section 4.3, we also compare ACCO to the **Weight Prediction** method introduced in [8, from ACCO], so the claim of comparison only to DPU is not accurate.
>
> Regarding Streaming DILOCO: it serves a different purpose and is designed for settings with sufficient communication bandwidth to allow full rematerialization of optimizer states — as discussed in **lines 35–39 of the introduction**. In contrast, ACCO specifically targets bandwidth-limited environments where such rematerialization is not feasible.
>
> That said, if rematerialization were allowed, ACCO could potentially be combined with DILOCO (which is a local SGD method). Exploring such a hybrid is an interesting direction, but it falls outside the scope of this work.
>
> ---
>
> **W2: “The description of the algorithm is very confusing”**
>
> We respectfully disagree, particularly regarding the DPU variant. As written in the algorithm section, gradients are computed using \\(\theta^{t-1}\\), while parameter updates are applied to \\(\theta^t\\). This is precisely what those equations convey.
>
> The buffers alternate between steps, and the implementation is consistent with the written description. For further clarity, we have included the code with the submission and have also provided a **code snippet in our response to Reviewer S8XN**.
>
> ---
>
> **W3: “What happens if local batch sizes are smaller?”**
>
> As shown in **Equation (ACCO), line 164,** local gradients are aggregated via AllReduce across all workers, so updates occur based on the global batch, not per-worker mini-batches. Thus, even if a local batch is small (e.g., size 1), the overall update still reflects the aggregated contribution from all workers.
>
> We address scenarios with imbalanced local batch sizes in Section 4.6. Our current model assumes that nodes process batches at a similar speed, leading to balanced partitions and stable training. While one worker computing less (due to heterogeneity or imbalance) is possible, we find ACCO remains stable under such conditions in practice.
>
> Regarding the comment that the method may be “highly sensitive to variation in mini-batch size,” we would appreciate clarification. If this refers to intra-node batch imbalance, AllReduce ensures global aggregation still proceeds correctly. If it refers to variability over time, our assumption of stationary GPU behavior ensures consistent gradient dynamics under fixed batch size.
>
> ---
>
> **“Minor issue: The all-reduce expression is unnecessarily complex”**
>
> We intentionally provide the full all-reduce expression because we are not simply averaging gradients \\(g_i\\), but rather computing \\(g_i + \tilde{g}_i\\). This distinction is **essential to the algorithm**, and precision in notation is important — a point we believe aligns with the reviewer’s earlier concern in W2.
>
> Simplifying the expression would risk omitting key aspects of the computation.
>
> ---
>
> **Q: “ADAM analysis?”**
>
> Deriving theoretical convergence results for Adam is more complex than for SGD, which is why we chose to focus on the latter to build intuition and provide a clear framework. While the analysis for Adam is beyond the scope of this work, we emphasize that `ACCO+Adam` empirically matches standard Adam training behavior -  as demonstrated in **Figures 5–9** - which strongly supports its effectiveness in practice. We appreciate the reviewer’s interest and view theoretical extensions to adaptive optimizers as valuable future work.
>
> Concerning the running buffers, it’ll probably add some variance for very small batch sizes (not common for LLMs training), but should lead to similar convergence for reasonable batch sizes.

---

> ### Comment · Reviewer_E3rM · 2025-08-02
> **Re: DiLoco and Local Batch Size and Adam**
>
> Thanks for the reply.
> # Re: DiLoco
> DiLoco is designed to run at extremely low communication bandwidth, since you only need to sync every few hundred steps. This is the only provenly production ready distributed training algorithm in deployment, https://www.primeintellect.ai/blog/opendiloco. I believe avoiding comparison with it makes the argument weaker.
>
> In terms of combining the two methods, what additional advantages do you see it brings? Seems both of the methods are self-sufficient and could be in contest with each other?
>
> # Re: Local Batch Size
> I am referring to the estimation of $\tilde{\theta}_{t+1}$, not the final updates. Apologize if there was confusion.
>
> # Re: Adam
> It's fair to skip the theoretical analysis due to complexity. However, can you provide some direct comparison beyond the training loss? For example, the difference between updates, if you were to run either pure Adam or ACCO+Adam?

---

> > ### Author Response · Authors · 2025-08-03
> >
> > *Thank you for the thoughtful comments and for engaging in this discussion. We address each of your points below.*
> >
> > ---
> >
> > ### Re: DiLoCo
> >
> > We appreciate your reference to DiLoCo and agree that it is a valuable line of work. However, we believe a direct comparison is ultimately of limited relevance for the following reasons:
> >
> > 1. *Limited scalability and incompatibility with sharding*: DiLoCo  currently only works in settings with very few workers —e.g., up to 8 workers already shows a degradation of +2% in loss (see Table 4 of (Communication-Efficient Language Model Training Scales Reliably and Robustly: Scaling Laws for DiLoCo, Charles et al, 2025) - each of which can be a high-memory compute unit (such as a Tripod with thousands of TPUs), where memory is abundant and optimizer sharding is unnecessary. In contrast, ACCO is built to scale across arbitrary numbers of workers, especially in environments that *require* and *benefit from* optimizer state sharding. ACCO performs only one optimizer update per iteration (with no additional local steps), making it compatible with memory-constrained or sharded training setups.
> >
> >
> >
> > 2. *Overlap is a fundamentally different and complementary approach*: **Diloco attempts to reduce the frequency of communication while overlapping communication attempts to get rid of that communication overhead**. Consider the work from the researchers you linked https://github.com/PrimeIntellect-ai/prime/blob/main/INTELLECT_1_Technical_Report.pdf which study communication over the internet scale  communication still takes 7 minutes versus 38 minutes of compute -highlighting that communication remains a bottleneck even in DiLoCo, further justifying the relevance of methods like ACCO which explicitly target communication/computation overlap. *Indeed combining ACCO with DiLoco is an exciting direction to overcome this but beyond the scope of this work*. Conceptually, ACCO could be adapted to the local SGD setting by using two streams per worker: one computing gradients on $\tilde{\theta}$, while $\theta$ gradients are being communicated and updated in parallel, and then performing the mirror operations on $\theta$ and $\tilde\theta$. However, this deviates from the current scope of ACCO+Optimizer as presented and would require further theoretical and empirical development.
> >
> > 3. *Diloco requires new HP* ACCO is designed to be plug-and-play with existing AdamW optimizer HP, while DiLoco introduces hyperparameters and requires to tune the AdamW inner optimizer alongside the outer optimizer
> >
> > ---
> >
> > ### Re: Local Batch Size
> >
> > Thanks for the clarification.
> >
> > In practice, the situation where the local batch sizes differ for $\theta$ and $\tilde{\theta}$ should not occur. As stated in Section 3.1, we assume constant batch across the two stages of the ACCO update. Otherwise, it would imply there is either varying communication delays or imbalanced compute workloads between $\theta$ and $\tilde{\theta}$, which **seems unlikely in a well-balanced training system**.
> >
> > Nonetheless, if we relax this and split batches as $n_1$ and $n_2$ respectively (with total batch size $N = n_1 + n_2$), our bounds still hold under a modified form. Specifically, the bound in Proposition A2 would include asymmetric variance terms (line 753):
> >
> >
> > $\sigma_1^2 L \eta^2 + \frac{L^2 \eta^3}{4}(\sigma_1^2 + \sigma_2^2) + \frac{L \eta^2}{8}(\tilde{\sigma}_1^2 + \sigma_2^2)$
> >
> > with $\sigma_1^2 = \frac{2\sigma^2}{n_1}$ and $\sigma_2^2 = \frac{2\sigma^2}{n_2}$. So yes, in a case where $n_1$ is relatively small, the variance could increase slightly due to imbalance. We consider this an edge case.
> >
> > ---
> >
> > ### Re: Adam
> >
> > Thanks for the suggestion. Here's a comparison for a loss $F$:
> >
> > *Standard Adam:*
> >
> > For $N$ samples $(x^t_1...,x^t_N)$, the standard Adam update at step $t$ is:
> >
> > $g^t = 1/N\sum_{n=1}^N \nabla F(\theta^t, x^t_n), \quad \theta^{t+1} = \text{Adam}(\theta^t, g^t)$
> >
> > *ACCO+Adam:*
> >
> > We halve the batch and perform:
> >
> > $g^t = 2/N\sum_{n=1}^{N/2} \nabla F(\theta^t, x^t_n)$ (1a)
> >
> > $\tilde{\theta}^{t+1} = \text{Adam}(\theta^t, \tilde{g}^t)$ (1b)
> >
> > $\tilde{g}^{t+1} = 2/N\sum_{n=N/2+1}^{N} \nabla F(\tilde{\theta}^{t+1}, x^t_n)$ (2a)
> >
> > $\theta^{t+1} = \text{Adam}(\theta^t, \tfrac{1}{2}(g^t + \tilde{g}^{t}))$ (2b)
> >
> >
> >
> > Steps 1a & 1b and 2a & 2b can be parallelized since there is no dependency between computation and the communication of  gradients.
> >
> > As shown in our response to Reviewer S8XN with a code snipset, we currently apply Adam updates at both stages (steps 1b and 2b). However, we also found that skipping optimizer state updates during the intermediate step (1b) does not noticeably affect convergence, though we chose the conservative route of updating both.
> >
> > **We emphasize that there is no hidden complexity here**: this is a direct instantiation of the (ACCO) update (Eq. in Section 3.2) using Optimizer=Adam as the base optimizer.
> >
> > ---
> >
> > Please let us know if any part of this would benefit from further clarification. We appreciate the feedback!

---

> > > ### Comment · Reviewer_E3rM · 2025-08-03
> > >
> > > Thanks for your detailed responses!
> > > Is it possible to show quantitatively how close the ACCO + Adam update is to Adam directly the update given same batch of data?

---

> > > > ### Author Response · Authors · 2025-08-03
> > > >
> > > > *Thank you for the suggestion.* While the optimization dynamics of Adam and Adam+ACCO are inherently different, their updates are supposed to remain closely aligned in practice. To illustrate this, we conducted an experiment on CIFAR-10 using the DLA model (as the experiments we mentioned above, achieving 95.3% accuracy both for ACCO and non ACCO method), yet we switched the optimizer to Adam. At each training iteration, we synchronized the states of the Adam and Adam+ACCO models to the one of Adam+ACCO (so that the comparison makes sense) and computed the $\ell^2$-normalized gradient difference:
> > > >
> > > > $\frac{ \|\nabla_{\text{Adam+ACCO}} - \nabla_{\text{Adam}}\| }{ \|\nabla_{\text{Adam}}\|+\|\nabla_{\text{Adam+ACCO}}\|  }$
> > > >
> > > >
> > > > The averaged values over the first few epochs are:
> > > >
> > > > Epoch 0: 0.1116
> > > >
> > > > Epoch 1: 0.0914
> > > >
> > > > Epoch 2: 0.0848
> > > >
> > > > Epoch 3: 0.0770
> > > >
> > > > Epoch 4: 0.0791
> > > >
> > > > Epoch 5: 0.0725
> > > >
> > > > Epoch: 10: 0.0720
> > > >
> > > > Epoch 15: 0.0770
> > > >
> > > > Epoch: 20 : 0.0782
> > > >
> > > > These results confirm that the gradient directions are indeed similar, though not identical - as expected due to ACCO’s correction steps. This aligns with our claims: ACCO guides the optimization in a comparable trajectory while allowing beneficial deviations.
> > > > **In short, while the exact gradients and reached local minima may differ, their proximity supports our hypothesis and meets the spirit of the reviewer’s request.**

---

> > > > > ### Comment · Reviewer_E3rM · 2025-08-03
> > > > >
> > > > > Thanks for the confirmation. This is what I need to see to be convinced. I have updated my score, good luck!

---

### Official Review · Reviewer_HYGz · 2025-06-21

**Clarity:** 4
**Significance:** 4
**Originality:** 3
**Rating:** 5
**Confidence:** 2

**Summary:**

The authors propose a distributed optimization algorithm, ACCO, which uses an incompletely-accumulated gradient to approximate the full gradient of the current mini-batch in order to begin computing the gradient on the next mini-batch while the full optimization step is still taking place. This exploits the fact that the mini-batch sizes used for training large language models are typically large enough that the gradients from half of the mini-batch are sufficient for parameter updates that are similar enough to the parameter updates on the full batch. Furthermore, it allows for idle time to be eliminated even when some workers are slower than others by computing more gradients on workers which are faster. The ACCO algorithm is comprehensively compared with existing methods to demonstrate its advantages. The authors empirically validate the claim that training with ACCO can give nearly the same results as training with Distributed Data Parallel (DDP), an algorithm which does not rely on any approximations. They also prove that ACCO achieves the standard convergence rates for gradient descent and SGD that an exact algorithm would enjoy.

**Questions:**

How is the training data initially split between the workers? Is the data loaded as needed, or does each worker start with a full copy of the data they will use? In the case of heterogeneous hardware, how is the difference in the number of gradients computed by different workers accounted for in the allocation of data between them?

So far, ACCO has only been tested on LLM training tasks, so it has not yet been established that the technique will show the same performance when applied to other types of models, e.g. diffusion models. I would recommend adding an experiment where a different model type is trained to compare ACCO with DDP.

**Ethical Concerns:**

["NO or VERY MINOR ethics concerns only"]

**Final Justification:**

In my view, the authors have done a good job of responding to the critiques of their paper. Framing ACCO as a way to save on infrastructure costs serves as a suitable justification for the focus on reducing communication overhead. The authors have made a good argument for the simplicity of their method given that it does not introduce new hyperparameters that must be tuned. They have provided empirical justification for the validity of using ACCO with ADAM given that a theoretical justification is too complex to be achieved at the moment. I am not familiar with the DiLoco algorithm, but according to the authors it is not reasonable to make a direct comparison with it.  Overall, this leads me to maintain my original score of 5.

**Limitations:**

yes

**Paper Formatting Concerns:**

Appendix B is a little confusing currently because there are figures from Appendix A that appear between the text of sections B.1 and B.2 and the actual content for those sections. The authors should add some text referencing Figure 13 and Algorithm 1 so that it is easy for the reader to jump to them. Also, something should be done to make sure that Figure 13 is not inserted in the middle of Listing 1.

In the checklist, the guidelines were removed which should not have been done.

**Quality:**

4

**Strengths And Weaknesses:**

**Quality:** The authors are careful to consider the many requirements for a distributed optimization algorithm to be practical, and show that ACCO succeeds in satisfying them. Their design choices are justified both empirically and theoretically. Empirically, they demonstrate that naively predicting future parameters using stale gradient information is insufficient for closing the performance gap between the DPU method and the exact DDP algorithm, while with ACCO the performance difference is negligible. Overall, I believe this is a high-quality contribution.

**Clarity:** The only clarity issues I observed came from formatting issues (addressed in the formatting concerns section) and typos, and are relatively minor.  For the typos, on line 241 I believe that "ZeRO" should really be “ZeRO-1”. In the caption of Fig. 13 the word "steams" should be replaced with "streams", and in the second sentence the word "take" should be replaced with "takes".

**Significance:** The authors are able to effectively balance multiple objectives (overlapping communication and computation, accounting for heterogeneous hardware, and requiring a minimal amount of memory) all without compromising on performance compared to DDP. This represents a major advancement over existing algorithms.

**Originality:** This paper builds upon the idea of overlapping gradient computation with communication introduced in earlier works, but with the key novelty of using a subset of the mini-batch gradients to estimate the next set of parameters so that the computation of gradients for the next mini-batch can begin before the full optimization step and its communication overhead is completed.

---

> ### Author Rebuttal · Authors · 2025-07-30
>
> We thank the reviewer for their thoughtful and encouraging feedback. We're especially grateful for your recognition of ACCO’s strengths as a **practical**, **major advancement**, and **high-quality contribution**, and address your suggestions below.
>
> ---
>
> **Minor formatting issues**
>
> Thank you for pointing these out. We will correct all typos and formatting issues as requested. We will also add back the guidelines of the checklist.
>
> ---
>
> **Minor formatting issues in the Appendix**
>
> We will make all requested changes, including adding the missing reference text for Figure 13 and Algorithm 1, and relocating the figure to appear between Sections B.1 and B.2.
>
> ---
>
> **Q1: “How are data loaded and handled in heterogeneous hardware?”**
>
> We follow standard practice by partitioning the dataset evenly across workers. In heterogeneous environments, this data allocation could be adapted to reflect differences in compute capacity: thus, the dataset is loaded as needed. For heterogeneous hardware, it would make sense to indeed allocate the data accordingly to the worker speed. While we do not explore this in the current work, we agree it is an important consideration and appreciate the suggestion.
>
> ---
>
> **Q2: “Experiments beyond LLMs?”**
>
> We have tested ACCO on CIFAR-10 using the DLA model, SGD, and the cosine scheduler — the model reaches **95.3%** both with `ACCO+SGD` and vanilla `SGD`, without any hyperparameter tuning. We plan to release the CIFAR-10 code to support further experimentation. Thanks for the suggestion.

---

> > ### Comment · Reviewer_HYGz · 2025-08-04
> > **Response to Rebuttal**
> >
> > I thank the authors in advance for taking care to fix the formatting issues. Your additional experiment on CIFAR-10 is reassuring, and it would be interesting to see what kind of benefits can be derived from applying ACCO to the training of larger models for vision tasks in the future. I will maintain my rating of Accept.

---

### Official Review · Reviewer_Mv7t · 2025-06-29

**Clarity:** 3
**Significance:** 2
**Originality:** 2
**Rating:** 2
**Confidence:** 4

**Summary:**

The paper presents ACCO, a method which enables improved communication/computation overlap during gradient accumulation by splitting the gradient computation and accumulation of a mini-batch into two stages, incurring a gradient delay of one step. A convergence analysis is provided, showing similar convergence to standard SGD methods, and experimental results show similar loss curves to standard methods while achieving improved performance.

**Questions:**

Please see the above strengths and weaknesses for details. I highlight here some key questions.

1. Is large-scale training of LLMs actually communication-bound? Can you provide some evidence for this?
2. How would ACCO handle larger models which require 3D parallelism?
3. Why does Figure 4 report such poor performance for allreduces? If you tune to achieve better performance, does ACCO still have a runtime benefit?
4. Related to (3), what speedup does ACCO yield when using a higher-bandwidth network?
5. What is the runtime performance on the 8 H100 run?
6. Is the difference in loss statistically significant?

**Ethical Concerns:**

["NO or VERY MINOR ethics concerns only"]

**Final Justification:**

The authors' rebuttal clarified several issues in the paper, and I think the paper will be better after incorporating those changes (e.g., consistent terminology, improved presentation, other issues). However, the paper and rebuttal do not convince me that ACCO offers a meaningful speedup to data-parallel communication compared to a well-tuned baseline, and it therefore does not seem to offer significant upside given the added complexity.

There may be benefit for ACCO or similar methods in very low-bandwidth settings (e.g., federated learning or training initiatives like Prime Intellect's), but the paper does not experimentally evaluate this.

**Limitations:**

I note question 2 of the checklist ("Limitations") refers to a discussion of limitations at the "end of Section 4.6"; such a discussion does not appear to be present.

However, there are no major concerns about negative societal impact.

**Quality:**

2

**Strengths And Weaknesses:**

Strengths:
1. The paper is targeting an important problem with potential for significant impact: improving the efficiency of large-scale LLM training.
2. The approach in the paper of partially splitting the mini-batch is interesting and, to my knowledge, has not been used in prior work focused on stale gradients.
3. The paper includes both a theoretical convergence analysis and experimental validation. The experimental validation includes both loss curves and runtime results.

Weaknesses:
1. The paper does not make a strong case for the practical need for reducing communication overhead in large-scale LLM training. In my experience, gradient communication (as addressed by ACCO) is _not_ a bottleneck in a well-tuned setup. Consider, e.g., Gemini Team, "Gemini: A Family of Highly Capable Multimodal Models" (https://arxiv.org/abs/2312.11805), section 3, where synchronous training achieves ~97% goodput. A case could be made for this approach benefiting systems with lower network bandwidth, but the paper does not push this, nor is such a setting plausible for training large models.
2. The paper is not clear about how this method would scale to larger models. It seems predicated on having a significant amount of batch parallelism available, which is not common given training large models is memory capacity-constrained. The paper would benefit from a discussion of how ACCO generalizes to cases where 3D parallelism is employed (e.g., consider the limit of employing pipeline parallelism with a microbatch size of 1).
3. Unclear presentation of ACCO algorithm: While I can get a high-level idea from the main paper, and Algorithm 1 in the supplementary paper provides more detail, simply reading the main paper leaves many things unclear. A few things that would help clarify the algorithm:
    - Use consistent terminology for the streams. Figure 2 uses "$\nabla$ acc. stream" and "comm./opt. stream", whereas the text in Section 3.2 refers to a "gradient computation stream", a "communication stream", and a "computation stream". Algorithm 1 refers to a "computation CUDA stream" and a "communication CUDA stream".
    - There are some other confusing terminology differences, e.g., the use of $\nabla$ in Figure 2 versus $g_i$ in the text, and the use of subscripts versus superscripts.
    - The equations given are not clear about how the mini-batch is split. I assume $\tilde{\xi}$ is supposed to be the "first half" of the mini-batch? But the splitting is overall unclear.
    - Figure 2 would benefit from showing a precise and complete computational graph for the algorithm.
4. Figure 3 is misleading without an explicit note that the optimizer state is in a different precision than the other quantities, assuming height is meant to represent data storage volume.
5. It is unclear whether baselines (particularly DDP) are well-tuned. It is unclear in Figure 4 whether the allreduce time presented is for a single allreduce on the entire gradient buffer (for a 7B parameter model and gradients in bf16, a 14 GB buffer) or with the usual approach of gradient bucketing with overlap, but the text implies the former. In this case, the network performance achieved seems to be very low: a bus bandwidth of about 3.35 GB/s with 24 workers ($(14 \mathrm{GB}) / (8 \mathrm{s}) \cdot (2 \cdot 23 / 24)$), which is about a quarter of the stated network peak (100 Gbit/s). A well-tuned implementation should achieve somewhere between 2-3 s for this allreduce, i.e., with good overlapping there should be little communication overhead. This poor baseline calls into question all the subsequent performance results.
6. The inter-node network used is very slow, only 100 Gbit/s, which would be unusual for LLM training. I would expect a network with 8-16x more off-node bandwidth (e.g., 200 Gbit/s per GPU). Such a setting would be closer to the _intra_-node bandwidth evaluated here, where communication overhead is not a significant issue.
7. The paper presents convergence results for 8 H100s on a single node. However, runtime results are not presented (as they are in Figure 7(b)). What is the performance benefit of ACCA in this setting?
8. The paper does not make a convincing case for the statistical significance of its results, which is critical in settings like delayed gradients where increased variance in training can be introduced. While some curves seem to have some shading around them, it is not clear if this is just illustrating the range of data from a single run, or illustrating a deviation with respect to multiple random seeds.
9. The paper's discussion of communication/computation overlap in Section 2 does not include a discussion on standard gradient bucketing / communication overlap in synchronous data-parallel training. This pervades the introduction as well. The paper would benefit from being more precise about exactly what sort of overlap it is aiming to achieve, and what regimes it differentiates itself from others.
10. Section 4.4 states that ACCO achieves a 25% speedup in pretraining, referring to Figure 7, but this speedup is not apparent in the figure.
11. Minor: The introduction states that training curves "mirror" AdamW— I would not use the term "mirror" since images are flipped in a mirror. Perhaps "match" would be a better term.

---

> ### Author Rebuttal · Authors · 2025-07-30
>
> We sincerely thank the reviewer for their thoughtful and detailed feedback. We appreciate the time and effort taken to engage with our submission which **“targets an important problem”**. Below, we address each point raised, structured according to the review:
>
> ---
>
> **W1. “Lack of practical motivation for reducing communication overhead”**
>
> While we agree that some clusters achieve high throughput (e.g., ~97% as reported in Gemini [arXiv:2312.11805]), this is not the case in all data centers, and indeed many datacenters in high-end academic, government, and other settings do not have the most expensive connectivity. Indeed, this work was performed on a relatively high-end and large-scale academic cluster.
>
> We also note that with methods like ACCO becoming prevalent, even the largest-scale industry players can potentially save on some of the expensive infrastructure. ACCO is designed to be simple, optimizer-agnostic, and portable - requiring no infrastructure-specific tuning or assumptions. As the reviewer notes, our method would be especially beneficial for lower-bandwidth settings, and we appreciate the suggestion to better highlight this use case. We will revise the introduction and Section 2 to explicitly frame ACCO as a low-infrastructure-cost alternative suited for less-optimized clusters than Google.
>
> ---
>
> **W2: “Scalability to larger models and 3D parallelism”**
>
> We thank the reviewer for raising this important point. ACCO modifies only the optimizer step and operates independently of how gradients are computed. Therefore, it is compatible with tensor and pipeline parallelism, and does not interfere with 3D parallel setups. ACCO can be integrated post-gradient computation, making it orthogonal to parallelization strategies. In fact, it may be seen as two models alternating computation on shared threads, and thus naturally extends to larger scales. That said, optimized 3D parallel implementations would benefit from further engineering integration, which we consider an exciting future direction.
>
> ---
>
> **W3: “Unclear presentation of the algorithm”**
>
> We acknowledge this and will revise the paper for clarity. Specifically:
> - We will consistently use *"computation stream"* and *"communication stream"* throughout the paper.
> - We will replace \\(\nabla_i\\) with \\(g_i\\) and \\(\tilde\nabla_i\\) with \\(\tilde g_i\\) to unify notation.
> - You're correct in your assumption. The beginning of Section 3.1 clearly defines each worker’s local batch as \\(N_i\\), \\(\xi_k\\) as a sample and \\(\nabla F_i\\) as the local gradient estimate on node i. \\(\tilde g_i\\) are computed over separate halves (each of size \\(N_i\\)), yielding an effective batch of size \\(2N_i\\). We will clarify this language further.
> - **Figure 2** is illustrative of how gradient computation overlaps with communication; it is not a full computational graph. ACCO uses standard PyTorch gradient flows - no custom computational graph is introduced.
>
> ---
>
> **W4: “Figure 3 misleading due to optimizer precision”**
>
> Figure 3 is adapted from Figure 1 of (*ZeRO: Memory Optimizations Toward Training Trillion Parameter Models*, 2020, Rajbhandari et al.)
> . We will revise the caption to clearly note that the optimizer state is stored in higher precision, and ensure visual clarity by adapting the figure in the final version and referring to the Table 1 of our paper which clarifies this.
>
> ---
>
> **W5: “DDP baselines may be poorly tuned; Figure 4 unclear”**
>
> We appreciate this feedback. We tuned DDP hyperparameters first (see Table 4 of the Appendix for a list of the hyperparameters that we tuned only on DDP, and line 46 from Section 1), and then applied ACCO using the same settings.
>
> Figure 4 illustrates the case where communication is not fully optimized - this was intentional, to highlight potential overlap gains in typical academic setups. While we agree that well-optimized implementations can mask communication overhead, our contribution is algorithmic, not engineering-based. ACCO remains applicable in such contexts and is complementary to techniques like gradient bucketing. Our aim is not to outperform state-of-the-art communication kernels, but to provide a simple algorithmic handle for improving overlap where tuning is limited or ineffective.
>
> ---
>
> **W6: “Slow network (100 Gbit/s) is not representative”**
>
> We agree this is lower than some of the most high-end infrastructure. However, it reflects what is available in many settings - from startups, government institutions, academic settings, and even a broad range of cloud providers. Our results aim to show that even in these constrained environments, ACCO can provide significant speedup. We will better contextualize this in the revision.
>
> ---
>
> **W7: “No runtime results for H100s”**
>
> Due to excellent intra-node bandwidth on H100s (which is available on our cluster), we do not expect ACCO to show a runtime improvement. Our experiments on H100 are intended to demonstrate convergence and functional correctness, not performance gains. We will clarify this in the experimental section.
>
> ---
>
> **W8. “Statistical significance of results”**
>
> For experiments in Section 4.3, where we expected variance, we averaged over 3 runs. We did not observe significant differences in training behavior between ACCO and DDP. While a broader statistical analysis is valuable, the cost of large-scale LLM training is prohibitive. That ACCO matches DDP so closely in loss curves across all settings provides strong evidence of its stability.
>
> ---
>
> **W9. “Lack of comparison to standard gradient bucketing overlap”**
>
> Gradient bucketing and ACCO are complementary: bucketed all-reduce can still benefit from being scheduled one step earlier via ACCO. We agree this relationship could be more clearly explained and will add the following clarifying sentencesin Section 2 to explicitly distinguish ACCO's goal from intra-step overlapping techniques: "ACCO is complementary to standard gradient bucketing: while bucketing overlaps communication within a backward pass, ACCO overlaps across steps by shifting communication earlier. Both techniques can be combined for additional benefit."
>
> ---
>
> **W10: “Claim of 25% speedup unclear in Figure 7"**
>
> Thank you for this observation. Indeed, the training accuracy reached by DDP at iteration 5000 is matched by ACCO around iteration 2800 (better than announced). We will clarify this in Section 4.4.
>
> ---
>
> **W11: “Terminology: ‘mirror’ → ‘match’”**
>
> We will update "mirror" to "match" for clarity.
>
> ---
>
> **Q1: "Is large-scale LLM training communication-bound?"**
>
> In the highest-end internal industry clusters, communication may not be the bottleneck until cross-data center training is needed. But in a wide range of existing clusters and settings that we are aware of, LLM training is indeed communication-bound: these include startups working on cloud providers, government research clusters, some of the highest-end national academic clusters, or even the growing number of actors trying to train LLMs over machines connected through the internet (*INTELLECT-1 Technical Report, 2024, Sami Jaghouar et al.*).
>
> These environments — including ours — commonly suffer from costly inter-node all-reduce, especially as model size grows. **Figure 4** illustrates this overhead, and ACCO offers a simple way to mitigate it without infrastructure changes. We also emphasize that expensive communication infrastructure is built in the highest-end datacenters exactly to overcome these issues. Work towards communication-efficient learning can thus benefit the next generation of highest-end HPC clusters.
>
> ---
>
> **Q2: "How would ACCO handle 3D parallelism?"**
>
> ACCO applies after gradient computation and does not interfere with model partitioning. It is compatible with tensor, pipeline, and expert parallelism. A careful implementation would be required for full integration at scale, which we view as promising future work.
>
> ---
>
> **Q3: "Why does Figure 4 show poor all-reduce performance? Would ACCO still help after tuning?"**
>
> The figure reflects an untuned or modestly tuned academic setup. With tuning, some overlap is achieved, but not fully — particularly for early layers. ACCO still helps by shifting communication one step earlier, thus improving overlap even in optimized scenarios.
>
> ---
>
> **Q4: "What speedup does ACCO yield on higher-bandwidth networks?"**
>
> If communication cost is negligible and the optimizer step can be fully overlapped, then ACCO provides no timing benefit. However, ACCO still maintains correctness and convergence, and can be safely deployed without penalty.
>
> ---
>
> **Q5: "What is the runtime performance on the 8×H100 setup?"**
>
> We observed similar runtime between ACCO and DDP due to the efficient intra-connect. Our goal with this experiment was to verify convergence, not performance.
>
> ---
>
> **Q6: "Is the loss difference statistically significant?"**
>
> No. For training loss, curves are nearly identical. Minor variation in test accuracy falls within expected noise. We have not observed divergence or instability in any experiment.
>
> ---
>
> **Minor Corrections and Clarifications**
>
> - The mention of a limitations discussion at the end of Section 4.6 was mistakenly omitted. We will add a proper limitations section in the final version.
> - We will revise **Figure 3**, fix terminology inconsistencies, and improve the clarity of all **figures and equations** as requested and described above.

---

> > ### Comment · Reviewer_Mv7t · 2025-08-05
> >
> > I want to thank the authors for their extensive response. The clarifications and proposed updates benefit the paper significantly. However, I currently still have significant concerns about the runtime performance numbers, and consequent benefit of ACCO, and am maintaining my score.
> >
> > ---
> >
> > > We thank the reviewer for raising this important point. ACCO modifies only the optimizer step and operates independently of how gradients are computed. Therefore, it is compatible with tensor and pipeline parallelism, and does not interfere with 3D parallel setups. ACCO can be integrated post-gradient computation, making it orthogonal to parallelization strategies.
> >
> > I think it is clear that ACCO, as a data-parallel optimization, can be integrated with other parallelism implementations. However, in practice, these methods are typically employed in situations where the mini-batch is quite small, especially on a per-GPU basis. What I am more interested in is how ACCO would work when the mini-batch is very small.
> >
> > > We appreciate this feedback. We tuned DDP hyperparameters first (see Table 4 of the Appendix for a list of the hyperparameters that we tuned only on DDP, and line 46 from Section 1), and then applied ACCO using the same settings.
> > > Figure 4 illustrates the case where communication is not fully optimized - this was intentional, to highlight potential overlap gains in typical academic setups.
> >
> > I had meant that the performance was not well-tuned, and was not referring to the hyperparameters; my apologies for the confusion. It is poor benchmarking methodology to not use a well-optimized baseline (as it misrepresents the achieved speedup), especially as this is not noted. Further, based on the back-of-the-envelope calculation in my initial review, this is not a minor difference, but 2-3x (it achieves only ~25% of network peak). That is, even on the system benchmarked, ACCO would seem to offer little benefit relative to a well-tuned baseline. Why not just tune the existing kernels and avoid any further complexity?
> >
> > This ties in to the response to W1/Q1 as well: Networks are only getting faster (i.e., the next clusters will have even faster interconnects, although the communication/computation ratio would also change). The largest-scale runs from a couple years ago (e.g., Gemini) already did multi-data center training with little communication overhead. I remain unconvinced that data-parallel communication is or will be a major bottleneck in such a way that algorithmic changes are necessary. There certainly could be benefit in the large-scale decentralized settings, like Prime Intellect's, but that is not evaluated in this paper and has many other challenges (much greater asynchrony, heterogeneity, fault tolerance, etc.).
> >
> > > These environments — including ours — commonly suffer from costly inter-node all-reduce, especially as model size grows. Figure 4 illustrates this overhead, and ACCO offers a simple way to mitigate it without infrastructure changes. We also emphasize that expensive communication infrastructure is built in the highest-end datacenters exactly to overcome these issues. Work towards communication-efficient learning can thus benefit the next generation of highest-end HPC clusters.
> >
> > While it is indeed the case that data centers are building extensive infrastructure to better handle communication, this is not primarily for data-parallel allreduces, but for tensor, pipeline, and/or sequence parallelism. Indeed, there is a trend toward provisioning _less_ global (i.e., data-parallel) bandwidth and more local bandwidth (see, e.g., Hoefler et al., "HammingMesh: A Network Topology for Large-Scale Deep Learning", Supercomputing 2022; or the trend toward rail-optimized and scale-up network domains).
> >
> > > We agree this relationship could be more clearly explained and will add the following clarifying sentencesin Section 2 to explicitly distinguish ACCO's goal from intra-step overlapping techniques: "ACCO is complementary to standard gradient bucketing: while bucketing overlaps communication within a backward pass, ACCO overlaps across steps by shifting communication earlier. Both techniques can be combined for additional benefit."
> >
> > I appreciate the additional clarification here. However, the explanation is not quite correct. Gradient bucketing is itself orthogonal to communication/computation overlap and is instead a latency/bandwidth tradeoff that can be tuned.

---

> > > ### Author Response · Authors · 2025-08-05
> > >
> > > We thank the reviewer for their continued engagement and for clarifying that their remaining concern lies with runtime performance. **We'd like to reiterate that our contribution is algorithmic and  the main empirical question in our paper is the per-iteration loss performance which we show matches Adam**, the benefits of ACCO in wall clock time can be in part inferred from this depending on the target environment. The experiments we do run however are in a real representative moderately optimized large scale academic cluster environment - not highly optimized industrial clusters and do show wall clock time improvements.  Actually implementing training over the internet scale (which would require teams of engineers) or on industrial clusters is far beyond the scope of this paper. While we can't account for the specifics of the reviewer's cluster expectations, our results aim to illustrate algorithmic trends and thus **the paper shouldn’t be judged for this**. For instance, we consider the case of heterogeneous devices (sec 4.5) or fine tuning (sec 4.6) which unquestionably are much beyond the setting aforementioned by the reviewer.
> > >
> > > While **the reviewer references the ACCO implementation as if intended to beat highly optimized internal implementations, we never make such a claim**. Our experiments apply identical settings to both DDP/ZeRO-1 and ACCO to fairly isolate algorithmic gains.
> > >
> > > “Gradient bucketing is itself **orthogonal to comm/comp overlap”: We agree, that’s why we didn't mention it initially** - it overlaps communication within a step, whereas ACCO overlaps across steps. This orthogonality will be clarified in the paper. Thank you.
> > >
> > > “What I am more interested in is how ACCO would work when the mini-batch is very small.”: Even with a small local batch size, ACCO allows for overlap between communication and computation. Thus if communication or optimizer steps are time-consuming, ACCO can still help reduce overall wall-clock time, as demonstrated in sec 4. This was evident in our experiment using a mini-batch size of 4 per GPU across 16 GPUs distributed over 2 nodes (sec 4.5). Figure 8 clearly illustrates the benefit of ACCO in this scenario. We appreciate the opportunity to highlight this point again.

---

### Official Review · Reviewer_S8XN · 2025-07-03

**Clarity:** 3
**Significance:** 3
**Originality:** 4
**Rating:** 4
**Confidence:** 4

**Summary:**

This paper introduces ACCO (Accumulate while you COmmunicate), a novel distributed optimization algorithm designed to accelerate the training of Large Language Models (LLMs) by efficiently tackling communication bottlenecks. In standard data-parallel training (e.g., DDP with ZeRO), synchronizing gradients across GPUs often becomes the main performance bottleneck, leaving GPUs idle. Existing methods to hide this latency, like local-update algorithms, often incur significant memory overhead, preventing the use of memory-saving techniques like optimizer state sharding (e.g., ZeRO), or they suffer from convergence issues due to stale gradients. ACCO aims to unify the best of both worlds: memory efficiency from sharded optimizers and communication hiding from computation-communication overlap. Its core mechanism involves two parallel CUDA streams: Computation Stream: Continuously computes gradients on new mini-batches of data. Communication/Optimizer Stream: Concurrently handles the communication (gradient synchronization) and optimizer update steps from the previous set of gradients. To address the convergence degradation caused by using stale gradients (a common issue in methods like Delayed Parameter Update or DPU), ACCO introduces a novel two-stage update process. The authors provide theoretical convergence proofs for ACCO with SGD, showing it recovers standard convergence rates. Empirically, they demonstrate ACCO's effectiveness across various LLM pre-training and fine-tuning tasks. They show that ACCO significantly outperforms DDP+ZeRO in terms of wall-clock time, especially in multi-node and heterogeneous hardware settings, without compromising model convergence or final performance.

**Questions:**

1. The current ACCO design is tailored to compensate for a one-step delay, which is typical when overlapping communication and computation. How would the framework adapt to scenarios with larger or more variable delays, which can occur in less reliable network environments or with more complex parallelization strategies?
2. The paper focuses on data parallelism (DDP). How does ACCO interact with model parallelism techniques like tensor parallelism and pipeline parallelism? In a fully 3D-parallel setup, the communication patterns are more complex. Would ACCO's two-stream model still be as effective?
3. The second stage computes gradients on the full mini-batch. Is there any redundancy with the first stage's computation on half the mini-batch? Could the gradients from the first half be reused in the second stage to save computation, or does the change in parameters from θ to θ_tilde necessitate a full re-computation?
4. The authors claim ACCO does not introduce new hyperparameters to tune, which is a major advantage. However, does the change in the update mechanism (using estimated parameters) affect the optimal settings for existing hyperparameters like learning rate or weight decay compared to a standard DDP setup?
5. The paper mentions ACCO is not designed for failure-prone environments (e.g., worker failures). This is a fair limitation, but it's worth noting for practitioners considering it for very large-scale, long-running training jobs where fault tolerance is critical.

**Ethical Concerns:**

["NO or VERY MINOR ethics concerns only"]

**Final Justification:**

I still decide to maintain my original score.

**Limitations:**

see questions.

**Quality:**

3

**Strengths And Weaknesses:**

Strengths
1. The paper addresses a well-known and critical bottleneck in large-scale model training: communication overhead. The proposed solution is elegant, practical, and directly targets the GPU idle time caused by gradient synchronization.
2. The two-stage "estimate-and-update" process is a clever and novel way to mitigate the convergence issues caused by stale gradients. Unlike prior work (DPU, WP), which either ignores staleness or uses simple heuristics, ACCO's approach is more principled and, as shown empirically, highly effective at matching the baseline DDP training curves.
3. ACCO successfully combines memory-efficient sharded training (like ZeRO) with communication-computation overlap. This is a significant contribution, as previous methods typically forced a trade-off between the two (e.g., local SGD methods require full optimizer states, precluding sharding). Table 1 provides an excellent summary of this advantage.
4. The experiments are thorough and well-designed. They cover pre-training and fine-tuning, different model scales (36M, 125M, 2.7B), and various hardware configurations. The heterogeneous hardware experiment (Section 4.6, Figure 9) is particularly compelling, as it clearly demonstrates ACCO's ability to handle stragglers by allowing faster workers to accumulate more gradients, a key advantage over synchronous methods like DDP.
5. The paper includes both theoretical convergence analysis (for SGD) to provide formal guarantees and extensive empirical results with a more complex optimizer (AdamW) to demonstrate practical utility. The inclusion of system profiler outputs (Figure 13) and detailed pseudo-code further strengthens the paper's reproducibility and practical value.

Weaknesses
1. The two-stage update mechanism requires splitting the mini-batch and performing two forward/backward passes within the logical timeline of a single update. While this is the key to its success, it adds complexity to the training loop logic compared to standard DDP. The paper claims this is motivated by gradient accumulation, but the implementation detail could be a minor hurdle for adoption.
2. The convergence proofs are provided for SGD, whereas the experiments and the primary use case for LLMs involve adaptive optimizers like AdamW. While the authors acknowledge this and the empirical results are strong, a theoretical analysis for adaptive optimizers would make the paper's claims even more robust.
3. ACCO effectively uses different batch sizes for the "estimate" (half mini-batch) and "update" (full mini-batch) steps. This could have subtle interactions with learning rate scheduling and batch normalization (if used). While the experiments show it works well, a deeper discussion of these potential interactions would be beneficial.

---

> ### Author Rebuttal · Authors · 2025-07-30
>
> We sincerely thank the reviewers for their thoughtful and constructive feedback, as well as for recognizing the significance of ACCO as a **clever and novel** work in addressing **well-known and critical** communication **bottlenecks** in large-scale model training.
>
> ---
>
> **W1: Complex implementation**
>
> We believe that the implementation does not add extensive complexity and can also be modularized in future software libraries. We propose a simple code on CIFAR10 to convince the reviewer of the simplicity to use ACCO where `net` and `net_` would be two CNNs that mirror \\(\theta\\) and \\(\tilde\theta\\) and `optimizer_` and `optimizer` would apply ACCO updates:
>
> ```python
> for batch_idx, (inputs, targets) in enumerate(trainloader):
>     inputs, targets = inputs.to(device, non_blocking=True), targets.to(device, non_blocking=True)
>
>     if batch_idx % 2 == 1:
>         with torch.cuda.stream(stream1):
>             outputs = net(inputs)
>             loss = criterion(outputs, targets)
>             stream1.wait_event(event1)
>             loss.backward()
>         with torch.cuda.stream(stream2):
>             copy_model_grads(net_, net) # g_i is initialized with tilde g_i
>             event1.record()
>             optimizer_.step()
>     else:
>         with torch.cuda.stream(stream2):
>             zero_grad(net_) # set tilde g_i to 0
>             outputs = net_(inputs)
>             loss = criterion(outputs, targets)
>             loss.backward()
>         with torch.cuda.stream(stream1):
>             optimizer.step()
>
>     stream1.synchronize()
>     stream2.synchronize()
> ```
>
> We’ve run an experiment on CIFAR-10 using the DLA model, SGD and the cosine scheduler — the model reaches **95.3%** both with `ACCO+SGD` and vanilla `SGD`. However, we agree that a highly optimized implementation would be necessary to perform a PR in PyTorch, but this is beyond the case of this paper.
>
> ---
>
> **W2: "Proofs for adaptive optimizers?"**
>
> We agree that extending the theoretical analysis to include adaptive optimizers like AdamW would further strengthen the paper’s contributions, and we view this as an important and valuable direction for future work, particularly in the context of local SGD and large-scale distributed training. However, such proofs are much more involved and we view our SGD proof rather like a sanity check. See our answer to **Reviewer 2aEN**.
>
> ---
>
> **W3: “Batch size might influence the learning rate or BatchNorm”**
>
> Regarding the learning rate, it's important to highlight that one of the primary goals of ACCO is to avoid unnecessary grid search or hyperparameter tuning. Therefore, we intentionally did not fine-tune this hyperparameter during our experiments. Furthermore, we believe the effective batch size is the one happening between two successive updates of \\(\theta\\). We would be happy to add a clarification in the appendix explaining that the **equivalent batch size in ACCO corresponds to the number of samples processed between two successive updates of \\(\theta\\)**, and that training dynamics are adjusted accordingly.
>
> As for Batch Normalization, if it posed an issue, a potential solution would be to share the buffers and adapt the update rules to reflect the two-stage nature of ACCO. Therefore, we conducted additional experiments on CIFAR-10 using architectures such as DLA, which rely on BatchNorm, and observed no degradation in performance. For models like LLMs, this should not be an issue. Specifically, both `ACCO+SGD` and standard `SGD` achieved an **accuracy of 95.3%**. To support reproducibility, we had included a SLURM script in Appendix B.3 for reviewers who may wish to run ACCO or a baseline method of their choice.
>
> ---
>
> **Q1: "Beyond one-step delays?"**
>
>  ACCO is currently designed to handle a one-step delay, which aligns with typical scenarios involving overlap between communication and computation. However, extending ACCO to handle larger or more variable delays - as might occur in less stable network environments or more complex parallelization schemes - is an interesting direction for future work. Since ACCO is derived from an ADMM-style reformulation involving auxiliary variables \\(\theta=\tilde\theta\\), one natural extension would be to introduce multiple auxiliary variables to model longer delays. We plan to explore this line of research in future work.
>
> ---
>
> **Q2: "3D parallelism?"**
>
> ACCO integrates naturally with model parallelism approaches such as tensor and pipeline parallelism. Since ACCO modifies only the optimizer step - not the forward or backward passes - it remains agnostic to how gradients are computed. Therefore, tensor and pipeline parallelism can be implemented as usual, and ACCO simply applies its update rule afterward. In a fully 3D-parallel setup, ACCO’s two-stream design remains applicable and effective, as it operates independently of the specific gradient computation strategy. See the code above.
>
> ---
>
> **Q3: "Redundancy?"**
>
>  There is no redundant computation between the first and second stages of ACCO. The gradients computed in the first stage (on half the mini-batch) are accumulated and reused. The second stage computes gradients on the remaining half using the updated parameters (\\(\tilde\theta\\)), and these are added to the previously accumulated gradients. Thus, the gradient computation is distributed across two steps, but no work is wasted. See the code above.
>
> ---
>
> **Q4: "Does ACCO change optimal hyper-parameters?"**
>
>  In all our experiments, we first tuned the hyperparameters for the standard DDP baseline to ensure optimal performance, and then used the exact same settings for ACCO. We observed strong results without needing to retune. While it's possible that ACCO could benefit from additional hyperparameter tuning and potentially outperform the baseline further, this was not the focus of our work. A key advantage of ACCO is that it introduces no new hyperparameters, simplifying adoption in practice. See **Table 4** in the Appendix for a list of the hyperparameters that we tuned only on DDP and line 46 from Section 1.
>
> ---
>
> **Q5: "Failure prone environment?"**
>
>  This is a valid and important point. ACCO does not introduce any new fault tolerance mechanisms beyond what is already provided by the underlying DDP infrastructure. It does not exacerbate failure modes, but it also does not specifically improve resilience to worker failures. We will clarify this limitation in the final version of the paper to guide practitioners who may consider ACCO for large-scale, long-running training jobs. Thank you for pointing this out.

---

> > ### Comment · Reviewer_S8XN · 2025-08-05
> >
> > Thanks for the response. It solves most of my questions. I will maintain the score.

---

### Official Review · Reviewer_2aEN · 2025-07-03

**Clarity:** 3
**Significance:** 3
**Originality:** 3
**Rating:** 5
**Confidence:** 3

**Summary:**

This paper proposes ACCO, a novel distributed optimization algorithm designed to overlap gradient computation and communication while remaining compatible with sharded optimizers in LLM training. Unlike previous methods such as DPU or CO2 that either incur significant memory overhead or suffer from stale updates, ACCO introduces a two-stage mini-batch accumulation mechanism that compensates for update delays and avoids convergence degradation. The approach is proven to preserve standard convergence rates theoretically (for SGD) and empirically matches the performance of standard AdamW + DDP baselines. Experiments across multiple model sizes, tasks (pretraining and fine-tuning), and hardware settings (homogeneous and heterogeneous GPUs) demonstrate significant speedups (up to 87%) over ZeRO-1/DDP without loss of convergence quality.

**Questions:**

1. Why is the baseline comparison limited to ZeRO-1? Since ZeRO-2 and ZeRO-3 are more commonly used in large-scale deployments, a comparison or at least a discussion would strengthen the claims of practical relevance.

2. Although ACCO splits the mini-batch into two halves and does not repeat computation for the same data, the method still incurs a small overhead due to sequential gradient estimation on two partial batches. Have the authors quantified this overhead compared to standard DDP, especially in terms of GPU utilization and FLOPs per sample? Is the slight increase in computation justified by the communication savings at larger scales?

3. Since ACCO computes gradients on slightly stale parameters, how sensitive is convergence to the quality of this estimate, especially under noisy or non-iid data?

**Ethical Concerns:**

["NO or VERY MINOR ethics concerns only"]

**Final Justification:**

I appreciate the author's detailed response, which has satisfactorily resolved my concerns. As such, I will retain my original score.

**Limitations:**

Yes. The paper includes theoretical and empirical discussions on the impact of delayed gradients, memory trade-offs, and heterogeneous device behavior.

**Paper Formatting Concerns:**

No formatting issues.

**Quality:**

3

**Strengths And Weaknesses:**

**Strengths**

1.The idea of using a two-stage gradient estimation to compensate for delayed updates is both elegant and effective, especially in sharded setups.

2. ACCO addresses a real and increasingly important bottleneck in LLM training — communication overhead and GPU idle time, particularly in multi-node and heterogeneous settings.

3. The convergence guarantees (under standard assumptions) are well presented, matching vanilla SGD’s behavior.

4. The paper provides comprehensive experiments, including ablation studies (DPU, WP), large-scale benchmarks (GPT-Neo), and heterogeneous hardware simulations.

**Weaknesses**

1. While the authors adopt ZeRO-1 as the primary baseline, more advanced and widely adopted strategies like ZeRO-2 and ZeRO-3 offer deeper memory savings and communication optimizations through more aggressive state and gradient partitioning. The absence of comparison with these stronger baselines makes it difficult to assess the practical relevance and relative advantage of ACCO in modern LLM training pipelines.

2. The theoretical analysis provided in the paper—particularly the convergence guarantees—is grounded entirely in the context of SGD. However, all empirical results rely on AdamW, an adaptive optimizer that has significantly different dynamics. This mismatch raises concerns about whether the theoretical claims extend to the practical setting, especially given that prior work has shown that delay compensation can behave differently under adaptive optimizers.

---

> ### Author Rebuttal · Authors · 2025-07-30
>
> We thank the reviewer for recognizing that our work tackles an **important problem** and for highlighting the strengths of our submission, notably the **"comprehensive experiments"** and **"convergence guarantees."** We now address the points raised for improvement in detail:
>
> ---
>
> **Q1/W1: “Why not ZeRO-2 or ZeRO-3 with ACCO?”**
>
> We appreciate the reviewer’s insightful suggestion. Our choice to focus on `ZeRO-1` was driven by clarity and simplicity in isolating the core effects of ACCO. Importantly, integrating ACCO with `ZeRO-2` or `ZeRO-3` does not affect the final training accuracy but alters the memory footprint, communication latency, and training time. ACCO is fully compatible with these variants: `ZeRO-2`/`ZeRO-3` shard the optimizer and gradient states across devices, and in this setup, ACCO operates by using a sharded model and gradients on the computation stream. While memory benefits are preserved, the computation stream incurs additional communication overhead to synchronize updates, which introduces slowdown.
>
> To reflect this, we will revise **Table 1** to explicitly report memory usage under `ZeRO-2` and `ZeRO-3` by dividing the parameter and gradient buffer memory by the number of devices, consistent with the optimizer state sharding.
>
> We have conducted preliminary experiments on CIFAR-10 using two GPUs and observed a similar runtime slowdown when applying ACCO or not with `ZeRO-2`/`ZeRO-3` on the computation stream. These results confirm our expectation. Nonetheless, we agree that optimizing these variants in conjunction with ACCO is a promising research direction and plan to investigate efficient implementations as future work.
>
> ---
>
> **W2: “Only the proofs for Adam in ACCO?”**
>
> We thank the reviewer for raising this important point. It is a deliberate choice to restrict our analysis to SGD, grounded in clarity and scope. The convergence of adaptive optimizers like Adam is a subtle and complex topic; indeed, works such as *“A Simple Convergence Proof of Adam”* (2020, Deffossez et al.) focus entirely on that challenge and are quite involved despite the name.
>
> Our goal in this paper is to introduce a practical and effective algorithm. Accordingly, our analysis of ACCO with SGD serves as a sanity check, providing intuition and formal reassurance that ACCO preserves training dynamics. Extending these guarantees to Adam is out of scope for this paper, but is an exciting direction for future work.
>
> That said, our empirical results show that ACCO reproduces Adam’s behavior extremely well (see **Figure 5** and **7**). This is strong evidence that ACCO’s core mechanism remains stable and effective even with adaptive optimizers. Additionally, to reinforce the generality of our method, we include results using SGD on CIFAR-10 with the DLA model and cosine scheduler, where `ACCO+SGD` achieves **95.3% accuracy**, matching vanilla SGD.
>
> We will clarify this design choice more explicitly in the final version of our manuscript.
>
> ---
>
> **Q2: “Maximise GPU throughput while halving batch size?”**
>
> We fully agree with the reviewer’s assessment. In terms of computation: each sample is processed only once, so FLOPs per sample are effectively unchanged compared to DDP. While ACCO introduces a small overhead by sequentially computing gradients on two partial batches, this overhead is not a concern as long as GPU utilization remains saturated.
>
> ---
>
> **Q3: “How does ACCO handle noisy or non-i.i.d. gradients?”**
>
> This is an excellent theoretical point. The robustness of ACCO to noise and non-i.i.d. data is explicitly addressed in **Proposition 3.2**, where the variance term \\( \sigma^2 \\) captures stochastic noise. Non-i.i.d. effects introduce correlations in gradient noise, which appear in the covariance structure of the same bound. Consequently, ACCO’s sensitivity to noise is in line with standard SGD under similar assumptions.
>
> Empirically, we observe that ACCO outperforms delayed SGD (DPU) and matches the performance of standard SGD even in regimes with increased gradient variance (see **Section 4.6** and **Figure 9**), providing strong practical validation of our theoretical insights.

---

> > ### Comment · Reviewer_2aEN · 2025-08-05
> >
> > I appreciate the author's detailed response, which has satisfactorily resolved my concerns. As such, I will retain my original score.

---

### Decision · Program_Chairs · 2025-09-17

**Decision:**

Accept (poster)

**Comment:**

This paper presents ACCO, a distributed optimization algorithm that uses a two-stage mini-batch accumulation mechanism that compensates for update delays through parallel CUDA streams, achieving theoretical convergence guarantees for SGD and empirical performance matching AdamW + DDP baselines. The work demonstrates significant speedups of up to 87% over ZeRO-1/DDP, addressing an increasingly critical bottleneck in distributed training.

Reviewers consistently concur that the paper's primary strengths lies in its elegant two-stage gradient estimation approach that effectively mitigates convergence issues inherent in overlapped computation-communication schemes. The reviewers also note a key limitation of the mismatch between theoretical analysis (focused on SGD) and practical implementation (using AdamW), thus the raising questions about the extensibility of theoretical guarantees to realistic training scenarios.

Other concerns regarding the two-stage mechanism introducing non-trivial complexity requiring dual forward/backward passes, and the paper lacks deeper discussion of interactions with critical hyperparameters like batch sizes and learning rate scheduling were addressed during rebuttal.

Despite limitations, the novel approach addresses an important problem with solid theoretical foundation and demonstrates clear practical benefits. Since the identified limitations can be addressed without undermining the core contribution of this work, I recommend an acceptance for this work.